# ELLMob: Event-Driven Human Mobility Generation with Self-Aligned LLM Framework

**Yusong Wang**[1*], **Chuang Yang**[2*], **Jiawei Wang**[2], **Xiaohang Xu**[2], **Jiayi Xu**[2], **Dongyuan Li**[2]
**Chuan Xiao**[3], **Renhe Jiang**[2†]
[1] Institute of Science Tokyo, [2] The University of Tokyo, [3] The University of Osaka
wangyi@lr.pi.titech.ac.jp, chuang.yang@csis.u-tokyo.ac.jp, {jiawei,xhxu}@g.ecc.u-tokyo.ac.jp
{xujy,lidy}@csis.u-tokyo.ac.jp, chuanx@ist.osaka-u.ac.jp, jiangrh@csis.u-tokyo.ac.jp

## Abstract

Human mobility generation aims to synthesize plausible trajectory data, which is widely used in urban system research. While Large Language Model-based methods excel at generating routine trajectories, they struggle to capture deviated mobility during large-scale societal events. This limitation stems from two critical gaps: (1) the absence of event-annotated mobility datasets for design and evaluation, and (2) the inability of current frameworks to reconcile competitions between users' habitual patterns and event-imposed constraints when making trajectory decisions. This work addresses these gaps with a twofold contribution. First, we construct the first event-annotated mobility dataset covering three major events: Typhoon Hagibis, COVID-19, and the Tokyo 2021 Olympics. Second, we propose ELLMob, a self-aligned LLM framework that first extracts competing rationales between habitual patterns and event constraints, based on Fuzzy-Trace Theory, and then iteratively aligns them to generate trajectories that are both habitually grounded and event-responsive. Extensive experiments show that ELLMob wins state-of-the-art baselines across all events, demonstrating its effectiveness. Our codes and datasets are available at https://github.com/deepkashiwa20/ELLMob.

## 1 Introduction

Human mobility generation aims to synthesize plausible spatio-temporal trajectories of human movement (Kim et al., 2024). The study of such trajectories offers deep insights for urban planning, transportation management, and public health (Duan et al., 2023; Chen et al., 2023; Li et al., 2024). Moreover, synthetic trajectories provide a privacy-preserving alternative that permits broader access and usage than sensitive real-world data. The emergence of Large Language Models (LLMs) has modeled trajectories as a "spatio-temporal language on a map," shifting the task from data distribution learning of traditional methods to instruction-based text generation (Choi et al., 2020; Feng et al., 2025). They leverage powerful contextual understanding and reasoning capabilities, offering advantages in semantic interpretability and versatility to different scenarios (Wang et al., 2024).

Current LLM-based methods explore various modeling strategies for generating realistic trajectories. One line of work studies single-stage direct prompting. For example, Wang et al. (2023); Feng et al. (2024) concatenate available information such as long and short-term check-ins and instruct an LLM to jointly model user preferences, geospatial distance, and sequential dynamics, yielding coherent trajectories. Profile augmented modeling with a multi-stage pipeline is also a prominent research direction. For instance, Wang et al. (2024); Gong et al. (2024); Ju et al. (2025) first apply an LLM to infer semantic profiles from user histories such as personas and travel motivations, and then condition generation on these high-level abstractions to produce personalized trajectories.

Although these methods achieve remarkable success, they still suffer from two key weaknesses. The first is **data scarcity leading to evaluation bias**. These methods are developed and evaluated primarily on datasets dominated by non-event days (stable period), resulting in questionable reliability

---

[*]Equal contribution.
[†]Corresponding author.

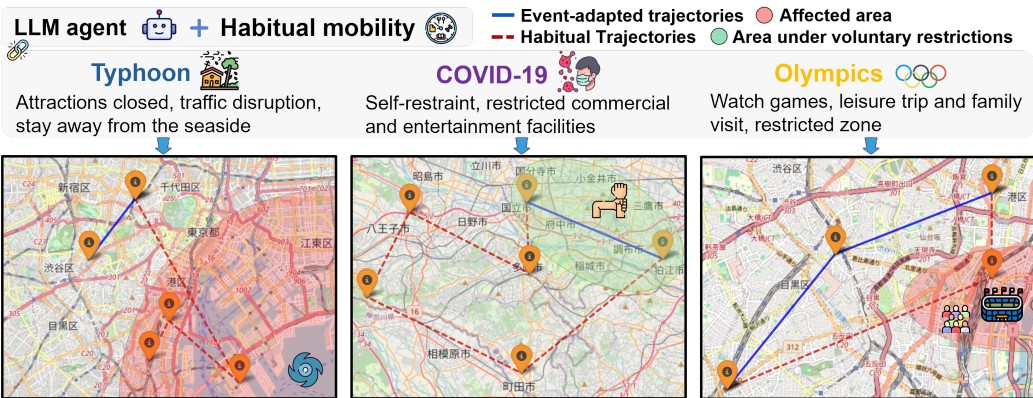

Figure 1: Event-driven mobility generation by LLMs, which incorporates event context to capture real-world human mobility on three different events: (1) Typhoon: evacuation from seaside, (2) COVID-19 Pandemic: self-restraint, and (3) Olympics: restricted zones and traffic jam.

when modeling non-routine deviations caused by large-scale societal events (e.g., natural disasters, public-health emergencies) (Zhong et al., 2024). As Figure 1 (a) illustrates, during a typhoon, travel shifts away from coastal areas and unnecessary commutes are eliminated. Without reliable data to evaluate their performance in modeling these dynamics, the reliability of these models for downstream applications, such as emergency response planning and transportation management under stress, is severely reduced (Li et al., 2017). Another limitation is **lack of a mechanism to reconcile competing decisions**. In these events, real-world human mobility combines habitual regularities with shock-induced deviation (Song et al., 2014). As Figure 1 (b) and (c) show, while overall mobility patterns are altered, event-adapted trajectories preserve visits to essential anchor points (shared nodes for both lines) of a user's routine, such as workplaces. Current methods struggle to navigate this duality, producing trajectories that either default to habitual patterns or are dominated by event constraints. Thus, an explicit reconciliation mechanism is needed to generate plausible trajectories.

To tackle these challenges, we first develop an event-centric dataset to provide the necessary empirical foundation for studying non-routine mobility. It covers trajectories from over a thousand users in the Tokyo metropolitan area cross three large-scale societal events (COVID-19 Pandemic, Typhoon Hagibis, Tokyo Olympics) with distinct mobility effects, in addition to a normal period for baseline comparison. Second, we introduce ELLMob, a self-aligned LLM framework that incorporates cognitive theory to shift generic self-alignment from error correction to conflict reconciliation, explicitly arbitrating between these competing decisions. Our key insight draws from Fuzzy-Trace Theory (FTT) (Reyna & Brainerd, 1995), which posits that *gist*, the essential meaning distilled from information, guides decisions under uncertainty. Event-driven mobility naturally fits this perspective, as individuals weigh habitual patterns against event-imposed constraints. Crucially, FTT reveals that gist can be linguistically expressed, enabling us to analyze the decision basis of LLMs. Building on these insights, ELLMob extracts three forms of gist to capture competing decision rationales: *pattern gist* (habitual tendencies) and *event gist* (constraint requirements), along with the *action gist* (LLM's current trajectory decision). Through iterative alignment of these gists, ELLMob transforms trajectory decision into a traceable process where competitions are explicitly identified and reconciled, generating habitually grounded and event-responsive trajectories.

Experiments show that on our event-centric dataset, existing methods produce trajectories that either default to routine patterns or overfit to event shocks (Figure 4), leading to poor generation quality. In contrast, ELLMob effectively reconciles this duality, achieving the best performance across four metrics, and surpasses strongest baselines by an average of **46.9%** across all three events (Table 5). Ablation studies confirm the critical role of gist-level reconciliation: incorporating cognitive-based self-alignment improves performance by an average of **69.5%** over non-aligned variants, highlighting its necessity for event-driven mobility modeling. Our contributions are summarized as follows.

- We construct the first event-centric human mobility dataset with detailed semantic information, providing a foundation for studying the non-routine deviations caused by societal events.

- We provide the first empirical evidence that current LLM-based methods struggle to model human mobility under societal events, revealing a critical research gap.

- We propose an FTT-inspired framework that constructs decision variables to explicitly reconcile conflicts between habitual patterns and event constraints, enabling traceable decision-making.

- ELLMob achieves state-of-the-art (SOTA) performance across all evaluated scenarios, demonstrating its effectiveness in generating plausible human mobility behaviors.

## 2 RELATED WORK

### 2.1 HUMAN MOBILITY GENERATION

The task of human mobility generation focuses on synthesizing realistic trajectories (Sun et al., 2023; Gong et al., 2023). Early deep learning methods applied sequential models like LSTMs and attention-based RNNs to predict temporal dependencies and personal preferences (Hochreiter & Schmidhuber, 1997; Kulkarni & Garbinato, 2017; Gao et al., 2017; 2018; Wang et al., 2018; Feng et al., 2018; Luo et al., 2021). To improve trajectory fidelity, subsequent research shifted to deep generative models, including VAEs (Huang et al., 2019), GANs (Choi et al., 2020; Wang et al., 2021; Zhao & Wang, 2023; Jia et al., 2024), and diffusion models (Zhu et al., 2023b;a; Chu et al., 2024), which excel at generating high-resolution location sequences. The emergence of LLMs introduced a new approach, re-framing trajectory generation as a sequence generation task conditioned on contextual prompts (Xue et al., 2022; Wang et al., 2023; Feng et al., 2024). However, the defined task for all these preceding models has been to simulate the routine activity trajectories of users. Their ability to generate faithful trajectories under sudden, non-stationary conditions such as disasters or public health crises therefore remains unknown, compromising their real-world application. Our work addresses this deficiency by defining the task of event-driven human mobility generation.

### 2.2 LLM FOR HUMAN MOBILITY MODELING

LLMs are currently applied across a range of human mobility modeling strategies (Wang et al., 2023; Feng et al., 2024; Wang et al., 2024; Tang et al., 2024; Zhang et al., 2024; Beneduce et al., 2025). For example, Wang et al. (2023) incorporated both long- and short-term dependencies from historical mobility data into LLMs to generate the next visiting location. Liang et al. (2024) used an LLM with historical mobility Origin-Destination data to generate travel demand during public events at the Barclays Center. Wang et al. (2024) integrated diverse user contexts, such as activity patterns, motivations, and profiles, into an LLM to generate more interpretable daily trajectories. Existing LLM-based methods fail to reconcile competing objectives during events: they either blindly follow habitual patterns or event constraints, making them unable to effectively adapt to sharp mobility behavioral changes driven by events (Luo et al., 2024). In contrast, ELLMob is cognitive theory-driven and incorporates a self-aligned mechanism that iteratively adjusts generated trajectories, shifting the generation goal from maximizing statistical likelihood to cognitive plausibility.

## 3 PROBLEM DEFINITION

In this section, we define the terminology and formulate the event-driven trajectory generation problem. A **trajectory** $\tau$ is a time-ordered sequence of visited places, represented as $\{(p_0, t_0), (p_1, t_1), \ldots, (p_n, t_n)\}$, where each tuple $(p_i, t_i)$ denotes a visit to place $p_i$ at time $t_i$. **Event Context**, denoted $E_{ctx}$, is structured data describing the exogenous shock associated with a specific event $c$. For a user $u$, we partition their historical trajectories within a pre-event window $W^{\mathrm{pre}}(c)$ into two disjoint sets based on a short-term duration $T_{\mathrm{short-term}}$ relative to the start time of event $t_c$: **long-term trajectories** $D_{\mathrm{long-term}}^{(u)} = \{\tau^{(u,d)} \mid d < t_c - T_{\mathrm{short-term}}\}$ and **short-term trajectories** $D_{\mathrm{short-term}}^{(u)} = \{\tau^{(u,d)} \mid d \geq t_c - T_{\mathrm{short-term}}\}$. These two datasets jointly characterize the prior mobility patterns of the user. The objective of this task is to develop a generative model that generates the event-driven trajectory: $F : (D_{\mathrm{long-term}}^{(u)}, D_{\mathrm{short-term}}^{(u)}, E_{ctx}) \mapsto \tau$.

## 4 EVENT HUMAN MOBILITY DATA

To develop and evaluate the performance of models in capturing mobility shifts under varying events, we construct a dataset from Tokyo trajectories collected via Twitter and Foursquare (2019-2021). It encompasses three distinct events selected to represent a spectrum of societal conditions, as well as a normal period to establish a baseline. Detailed specifications are provided in Table 1.

Table 1: Specifications for experimental evaluation scenarios.

| Event | Event Period | Pre-Event Period | Description |
|---|---|---|---|
| Typhoon Hagibis | 2019-10-12 ∼ 10-13 | 2019-08-13 ∼ 10-11 | Natural disaster. |
| COVID-19 Pandemic | 2020-04-07 ∼ 04-13 | 2020-02-07 ∼ 04-06 | Public health emergency. |
| Tokyo 2021 Olympics | 2021-07-23 ∼ 07-29 | 2021-05-24 ∼ 07-22 | Pandemic-era large event. |
| Normal Period | 2019-09-01 ∼ 09-30 | 2019-07-03 ∼ 08-31 | Regular urban mobility. |

For prolonged events (COVID-19 Pandemic and Tokyo Olympics), we focus on the first seven days to capture pronounced behavioral shifts. COVID-19 Pandemic began with the State of Emergency of Japan. 30-day window for normal period is used to establish a robust baseline of typical mobility, averaging out weekly fluctuations. Pre-event period (two months) acts as training data for deep learning baselines and a source for user pattern extraction for LLM-based baselines. Following collection process described in Appendix B, we curated a dataset of 1,100 users who exhibited consistently dense check-in activity throughout the study period. Each sample includes user ID, geographical coordinates, subcategory, subcategory ID, category, timestamp, and a comment, as shown in Appendix C. Table 2 shows key statistics of this dataset. Table 3 compares data dimensions across major mobility datasets (GeoLife (Zheng et al., 2009), Gowalla (Cho et al., 2011), Foursquare (Yang et al., 2015), and Yelp (Asghar, 2016)), revealing that ours cover all standard mobility dimensions. To the best of our knowledge, our dataset is the first to cover a broad spectrum of distinct event types (long-term vs. short-term, diverse semantics) with continuous, dense pre- and during-event trajectories, enabling the precise analysis of behavioral transitions during societal shifts.

Table 2: Dataset statistics by scenario, detailing the counts of check-ins, unique POIs, and POI categories, with per-scenario totals.

Table 3: T, L, C, TC, N, and E denote time, location, subcategory, comments, normal period and explicit event annotation, respectively.

| Event | Check-ins | POIs | Subcat. | Dataset | T | L (lat,lon) | C | TC | N | E |
|---|---|---|---|---|---|---|---|---|---|---|
| Typhoon Hagibis | 4,330 | 3,284 | 334 | GeoLife | ✓ | ✓ | ✗ | ✗ | ✓ | ✗ |
| COVID-19 Pandemic | 8,448 | 4,629 | 330 | Gowalla | ✓ | ✓ | ✗ | ✗ | ✓ | ✗ |
| Tokyo 2021 Olympics | 16,069 | 9,586 | 467 | Foursquare | ✓ | ✓ | ✓ | ✗ | ✓ | ✗ |
| Normal Period | 104,781 | 36,487 | 655 | Yelp | ✓ | ✓ | ✓ | ✓ | ✓ | ✗ |
| Agg. Pre-Event | 641,747 | 125,710 | 764 | **Ours** | ✓ | ✓ | ✓ | ✓ | ✓ | ✓ |

To quantitatively ground this study, we present a statistical analysis to reveal distinct impacts of each event on collective mobility. We adopt four widely-used metrics from the human mobility work (Pappalardo et al., 2015; Alessandretti et al., 2020): daily check-ins, capturing activity intensity, the radius of gyration and total travel distance, measuring spatial extent and volume, respectively, and the daily activity duration, quantifying the temporal span. Results are shown in Figure 2. Specifically, COVID-19 Pandemic and Typhoon Hagibis significantly suppressed the scope and frequency of movement. In contrast, Olympics reversed suppressive trend in activity of COVID-19 Pandemic.

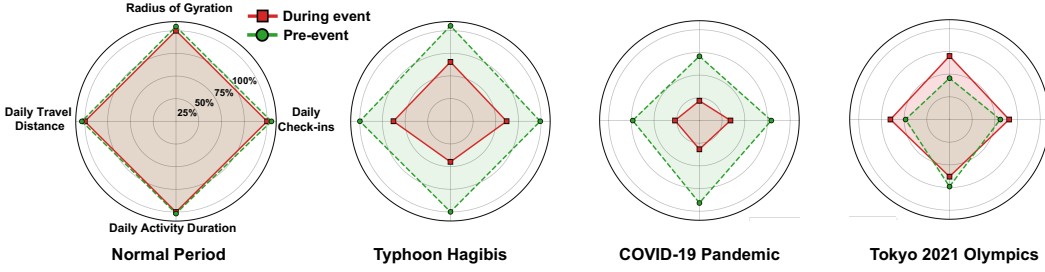

Figure 2: Normalized radar charts of four mobility metrics on pre-event and during-event patterns.

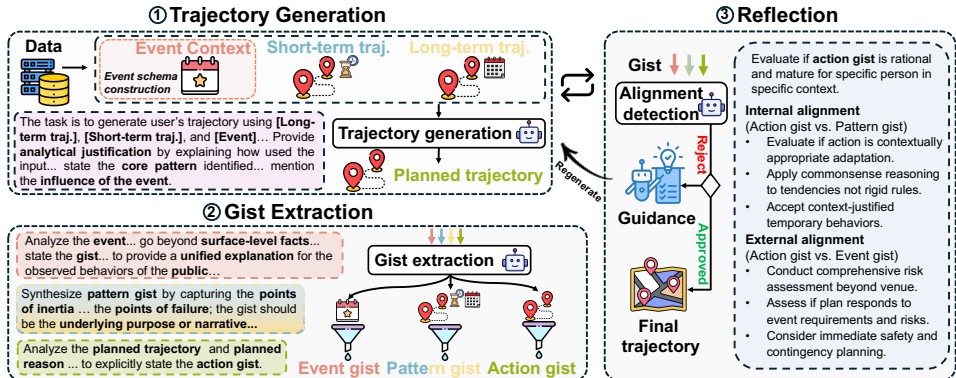

Figure 3: The ELLMob framework architecture comprising three interconnected modules: Trajectory Generation, Gist Extraction, and Reflection-based alignment.

## 5 METHODOLOGY

### 5.1 EVENT SCHEMA CONSTRUCTION

A challenge in event-driven mobility generation is that real-world events are typically described in lengthy, free-form text (e.g., news reports and policy documents), which often leads the LLM to overlook critical information during trajectory generation (Liu et al., 2024; An et al., 2024). To address this, an event schema construction step is introduced to transform raw event narratives into a structured representation that explicitly outlines the event's impact on population mobility patterns. The event schema is designed around four distinct but complementary aspects that collectively cover the key information required to assess mobility changes:

- *Event profile*: Records the fundamental elements of the event (e.g., type, name, occurrence time, and affected regions). This provides an anchor for spatio-temporal alignment.
- *Intensity and scale*: Quantifies key metrics of the event's severity, such as wind speed and amount of precipitation, to inform travel risk assessment.
- *Infrastructure and service impact*: Describes the operational status of critical resources (e.g., transportation, public venues), defining the physical constraints on mobility.
- *Official directives*: Captures governmental orders and recommendations (e.g., a request for residents to avoid non-essential travel), including their applicable populations and geographic scope, ensuring generated trajectories refer to policy mandates.

An LLM is leveraged to process the raw event text $E_c$ into a structured key-value format, referred to as event context $E_{ctx}$, which subsequently serves as the input to the trajectory generation task. The prompt is provided in Figure A6 and the generated contents are presented in Appendix E.

### 5.2 SELF-ALIGNED LLM FRAMEWORK

In Figure 3, we propose ELLMob, a cognitive theory-driven framework that employs an iterative refinement process to reconcile competition between a user's habitual patterns and event constraints.

#### 5.2.1 THEORY OF PLANNED MOBILITY BEHAVIOR

Fuzzy-Trace Theory (FTT) (Reyna & Brainerd, 1995) provides a cognitive perspective on decision-making under uncertainty, emphasizing that decisions are driven by *gist*, which refers to the bottom-line (essential) meaning of information rather than *verbatim* details. A classic example is evacuation, where the action is driven not by the exact probability that a tsunami will strike (e.g., 15%) but by the gist that the risk is "high." According to FTT, gist can be linguistically expressed, making the decision-making basis transparent. In event-driven mobility generation, uncertain disruptions such as natural disasters or epidemics require the model to navigate between two independent decision bases: adhering to habitual mobility routines or complying with event-imposed constraints. Existing LLM-based methods lack an explicit mechanism for arbitrating between these competing decisions, leaving their rationale difficult to audit or control and tending to follow only one gist. By extracting the gist underlying these decisions, we expose the model's decision basis and resolve competition in

a transparent manner. We extract relevant gists: *Pattern Gist* and *Event Gist*, corresponding to two independent decision bases, and *Action Gist*, representing LLM's tentative plan.

- *Pattern gist*: A representation of the essential tendencies distilled from the user's habitual mobility patterns, reflecting stable movement routines.
- *Event gist*: A representation of the essential tendencies distilled from contextual constraints, capturing constraints or incentives imposed by external events.
- *Action gist*: A representation of LLM's immature mobility decision, extracted from the candidate trajectory during planning.

We heuristically formalize these concepts as structured representations, where each gist is derived by assessing the relevant source data along a set of core attributes, which is illustrated in Table 4. Building on this, we propose a reflection module in which the LLM audits the alignment of these gist. This alignment process explicitly identifies conflicts that are then resolved through guided refinement to ensure that the final generated trajectory is grounded in a unified decision basis. Notably, FTT offers a architecture design basis. It motivates a multi-gist decision framework, guides mapping heterogeneous inputs into a unified gist space for consistent alignment, and drives the use of interpretable bottom-line attributes over arbitrary features. Ablation study is provided in Appendix F.

Table 4: A set of defined core attributes that guide the gist extraction from source information.

| Gist Type | Attribute | Description | Example |
|---|---|---|---|
| Pattern Gist | Core Behavior | The dominant pattern of action. | Daily commute to a office. |
| | Points of Inertia | Deeply embedded, non-negotiable components. | Returning home to a specific neighborhood at night. |
| | Points of Fracture | Critical dependencies and single points of failure. | Reliance on a single train line that might be suspended. |
| Event Gist | Primary Intent | Core implication of the event for mobility decisions. | High risk outdoors, strong incentive to stay home. |
| | Behavioral Implications | Survival, social dynamics, and compliance. | Evacuation from coastal areas, seeking indoor shelter. |
| | Risk-Reward Calculus | A cost-benefit analysis of the response to event risks. | Risk of injury outweighs reward of a non-essential outing. |
| Action Gist | Primary Intent | Main purpose driving this trajectory choice. | To get essential supplies from a nearby store. |
| | Habit Adherence | Degree of preservation in habitual patterns. | Low; this trip deviates from the usual work commute. |
| | Event Compliance | Trajectory's level of adherence to event constraints. | High; the trip is short and avoids dangerous areas. |

### 5.2.2 REFLECTION-BASED ALIGNMENT

We replace single-pass decoding with an iterative reflect-refine loop that externalizes the model's decision basis for transparent reasoning. Moreover, unlike generic self-alignment approaches that primarily correct errors such as hallucinations, our mechanism targets the decision-making dilemma inherent in event-driven mobility scenarios. Our alignment performs in two stages:

**Alignment Auditing.** This process is dedicated to rigorously auditing the plausibility of a planned trajectory. Each candidate trajectory is checked along two binary dimensions: *Internal alignment* is to ascertain whether the planned trajectory reflects a coherent expression of the user's intrinsic habitual mobility patterns and current behavioral tendencies. *External alignment* determines if the planned trajectory represents a rational and compliant response to the constraints and implications of the event. A trajectory is accepted only if both criteria are satisfied. The auditor outputs two binary judgments, accompanied by concise rationales that indicate any violated criterion and its cause.

**Corrective Refinement.** Should a planned trajectory fail to satisfy the criteria of either the internal or external alignment audit, ELLMob initiates a corrective refinement loop. During this loop, the precise reasons for the audit failure are provided as feedback to the trajectory generator, guiding it to regenerate a revised trajectory that explicitly addresses the identified semantic misalignments and logical flaws. This loop repeats up to a maximum of $K$ iterations. A trajectory that satisfies both criteria within the $K$-step budget is accepted as a final trajectory. In the rare event that constraints remain unmet after $K$ iterations, the system executes a fallback strategy. It accepts the last validated trajectory available in the agent's buffer and reports unmet constraints to ensure transparency.

To clearly understand the ELLMob, we show the overall procedure in pseudo-code form Appendix G with complete contents of all related prompts provided in Appendix M.

Table 5: Comparison of different methods under three events. Performance is evaluated by JSD across four dimensions with the best performance highlighted in **bold**.

| Models | Typhoon Hagibis | | | | COVID-19 Pandemic | | | | Tokyo 2021 Olympics | | | |
|---|---|---|---|---|---|---|---|---|---|---|---|---|
| | SI↓ | SD↓ | CD↓ | SGD↓ | SI↓ | SD↓ | CD↓ | SGD↓ | SI↓ | SD↓ | CD↓ | SGD↓ |
| LSTM | 0.1336 | 0.1039 | 0.0555 | 0.1111 | 0.1928 | 0.1047 | 0.1300 | 0.2571 | 0.1147 | 0.0651 | 0.0598 | 0.0634 |
| DeepMove | 0.1697 | 0.0826 | 0.0266 | 0.0759 | 0.1838 | 0.0834 | 0.0423 | 0.1688 | 0.1667 | 0.0492 | 0.0587 | 0.0555 |
| GETNext | 0.3031 | 0.2007 | 0.0274 | 0.1037 | 0.2891 | 0.2241 | 0.0142 | 0.1354 | 0.2701 | 0.1473 | 0.0176 | 0.1204 |
| MHSA | 0.1430 | 0.1815 | 0.0118 | 0.0711 | 0.2180 | 0.3083 | 0.0254 | 0.0437 | 0.1815 | 0.2013 | 0.0120 | 0.0525 |
| TrajGAIL | 0.1034 | 0.3591 | 0.0155 | 0.0275 | 0.1600 | 0.3557 | 0.0195 | 0.0444 | 0.0863 | 0.2913 | 0.0121 | 0.0104 |
| DiffTraj | 0.1271 | 0.2450 | 0.0385 | 0.0761 | 0.1405 | 0.2766 | 0.0554 | 0.0454 | 0.0732 | 0.2171 | 0.0342 | 0.0282 |
| LLMOB | 0.0949 | 0.1195 | 0.0123 | 0.0256 | 0.1013 | 0.1051 | 0.0186 | 0.0286 | 0.0973 | 0.0274 | 0.0110 | 0.0051 |
| LLM-MOB | 0.1214 | 0.0468 | 0.0285 | 0.0344 | 0.1166 | 0.0532 | 0.0234 | 0.0353 | 0.1047 | 0.0286 | 0.0085 | 0.0052 |
| LLM-Move | 0.1267 | 0.0392 | 0.0136 | 0.0303 | 0.1408 | 0.0567 | 0.0127 | 0.0503 | 0.1967 | 0.0298 | 0.0101 | 0.0057 |
| LLM-ZS | 0.1574 | 0.1348 | 0.0153 | 0.0724 | 0.1146 | 0.0576 | 0.0552 | 0.0570 | 0.0938 | 0.0330 | 0.0132 | 0.0052 |
| ELLMob | **0.0642** | **0.0200** | **0.0041** | **0.0173** | **0.1003** | **0.0444** | **0.0080** | **0.0268** | **0.0617** | **0.0061** | **0.0022** | **0.0035** |
| w/o I.A.&E.A. | 0.1304 | 0.1270 | 0.0139 | 0.0723 | 0.2331 | 0.1077 | 0.1190 | 0.0733 | 0.1465 | 0.0340 | 0.0093 | 0.0095 |
| w/o I.A. | 0.0835 | 0.0720 | 0.0135 | 0.0436 | 0.1235 | 0.0950 | 0.1053 | 0.0300 | 0.1355 | 0.0316 | 0.0088 | 0.0086 |
| w/o E.A. | 0.0680 | 0.0258 | 0.0077 | 0.0229 | 0.2237 | 0.0860 | 0.0283 | 0.0430 | 0.1392 | 0.0291 | 0.0083 | 0.0064 |
| w/o Eve. Ext. | 0.0736 | 0.0273 | 0.0045 | 0.0227 | 0.2037 | 0.0741 | 0.0269 | 0.0405 | 0.0686 | 0.0213 | 0.0030 | 0.0041 |

## 6 EXPERIMENTS

### 6.1 EXPERIMENTAL SETUP

**Baselines.** ELLMob is evaluated against two types of baselines: *1) deep learning-based methods* which include predictive models: LSTM (Hochreiter & Schmidhuber, 1997), DeepMove (Feng et al., 2018), GETNext (Yang et al., 2022), and MHSA (Hong et al., 2023); and generative models: TrajGAIL (Choi et al., 2020) and DiffTraj (Zhu et al., 2023b). *2) LLM-based models*: LLM-MOB (Wang et al., 2023), LLM-Move (Feng et al., 2024), LLMOB (Wang et al., 2024), LLM-ZS (Beneduce et al., 2025). For fairness, the input event information remains consistent across all LLM-based methods. Specifically, detailed event descriptions (including type, time, location, and constraints) are integrated as natural language context at the beginning of each prompt, ensuring that all baselines have equal access to the event information despite differences in their specific prompt designs.

**Evaluation Metrics.** *Step Interval (SI)*. The time between consecutive activities, defined as $\text{SI}_t = \tau_{t+1} - \tau_t$, where $\tau_t$ denotes the timestamp at step $t$; *Step Distance (SD)*. The distance between consecutive locations, defined as $\text{SD}_t = \|l_{t+1} - l_t\|_2$, where $l_t \in \mathbb{R}^2$ denotes the location at step $t$. *Category Distribution (CD)*. This metric captures the distribution of activity types. To calculate it, we aggregate the total number of visits $N(c_k)$ for each location category $c_k$. *Spatial Grid Distribution (SGD)*. It captures the population-level spatial footprint of activities. All visited locations are discretized onto a fixed $S \times S$ grid covering the Tokyo metropolitan area, with visit counts accumulated per grid cell. To mitigate sparsity, following Ouyang et al. (2018); Feng et al. (2020), the top $25\%$ frequently visited cells are retained for evaluation. For each of the four metrics, we form a distribution from the generated trajectories and compare it against the ground truth distribution using the Jensen-Shannon Divergence (JSD), following Zhu et al. (2023b); Wang et al. (2024).

**Implementation Details.** We primarily use GPT-4o-mini (2025-01-01-preview) (Achiam et al., 2023) as the backbone for its capability–cost balance, with additional LLM evaluations reported in Appendix H. Following Wang et al. (2024), we set the temperature to 0.1 to curb randomness, Top-p to 1, and model trajectories at a 10-minute resolution. Grid size parameter $S$ is set to 10. $K$ is set to 3 to balance refinement quality and inference cost based on the parameter study in Appendix K. A stability analysis verifying result consistency is provided in Appendix L.

### 6.2 QUANTITATIVE RESULTS

Table 5 summarizes our main results, demonstrating ELLMob's consistent superiority across all event-driven settings. For instance, it improves the SI score by 32.3% for Typhoon Hagibis and the SD score by 16.5% for the COVID-19 Pandemic compared to the strongest baselines. We observe that LLM-based approaches generally outperform traditional deep learning models, particularly on spatial coherence metrics (SD, SGD), benefiting from their ability to integrate event context. Furthermore, to verify spatial generalizability beyond the Tokyo area, we extended evaluations to Osaka during the COVID-19 pandemic, with detailed results provided in Appendix I.

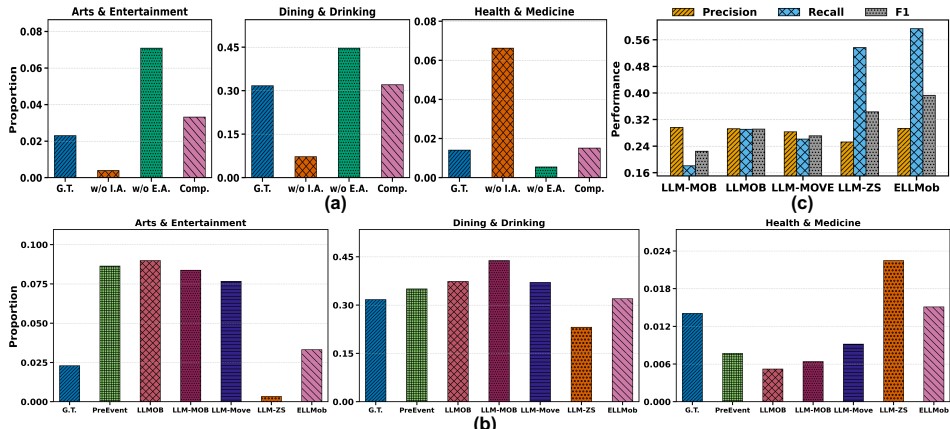

Figure 4: (a) Comparison of generated activity proportions (relative to the total number of activities) during the COVID-19 Pandemic. Each chart contrasts the ground truth (G.T.), distribution with: Without internal alignment (w/o I.A.), without external alignment (w/o E.A.), and the complete model (Comp.). (b) Comparison of three key activity categories distributions generated by ELLMob with various LLM-based baselines. (c) Performance comparison on the active user prediction task.

The ablation study further dissects our ELLMob's performance. Removing either the reflection module (w/o I.A.&E.A.) or the event schema (w/o Eve. Ext.) consistently degrades performance. The two components of the reflection module show distinct roles: Internal alignment (w/o E.A.) provides foundational plausibility and external alignment (w/o I.A.) acts as a scenario-specific corrective. The importance of external alignment is particularly evident during the COVID-19 Pandemic, where its removal causes a catastrophic 132.4% performance degradation in the SI score. This highlights its critical role in aligning with rational behaviors that substantially deviate from habitual patterns.

## 6.3 MODEL ANALYSIS

**Analysis of Self-alignment.** To dissect the distinct roles of internal and external alignment, we analyze the generated distributions of three sensitive top-categories (Arts & Entertainment, Dining & Drinking, and Health & Medicine) within the COVID-19 Pandemic. As shown in Figure 4 (a), removing either alignment leads to distinct failures. Lacking internal alignment, the model over-corrects for the event, generating an unrealistic surge in Health & Medicine while excessively suppressing Arts & Entertainment and Dining & Drinking activities. Conversely, without external alignment, the model rigidly adheres to habitual patterns, producing the opposite failure. Furthermore, Figure 4 (b) reveals that most LLM-based baselines default to habitual patterns (PreEvent) such as overestimating entertainment/dining while ignoring health-related travel, while LLM-ZS overcorrects by suppressing social activities entirely. Both extremes demonstrate that these baselines are unable to reconcile habitual patterns with event constraints in trajectory decision. The complete ELLMob successfully considers these two forces to produce a distribution that closely matches the ground truth, demonstrating the effectiveness of the self-alignment mechanism.

**Fundamental Decisions in Disasters.** Accurately identifying individuals who travel during extreme weather is critical for targeted early warnings and effective emergency response. We frame this as a binary classification task to identify a potentially high-risk cohort, which we define as the positive class of "active" users (at least one trip) during the typhoon. We evaluate LLM-based baselines for their strong ability to incorporate event context. As shown in Figure 4 (c), ELLMob achieves the highest F1-Score, driven by its superior recall of 59.3% in identifying this "active" high-risk population. This effectiveness is likely attributed to the iterative alignment process, which enhances LLM's joint understanding of individual user mobility patterns and event constraints.

**Case Study.** To illustrate ELLMob's reasoning process, Figure 5 presents a case study of a user with a strong culinary exploration pattern during the COVID-19 Pandemic. The initially planned trajectory (stay at home) over-aligns with the event's public health constraints and is flagged by our reflection module for conflicting with the user's habitual patterns. Guided by this internal feedback, the model iteratively refines the plan into a plausible trajectory that weighs both factors, ultimately limiting rather than eliminating dining outings. This case highlights the ELLMob's ability to reconcile user patterns with event constraints to generate realistic behaviors that match the ground truth.

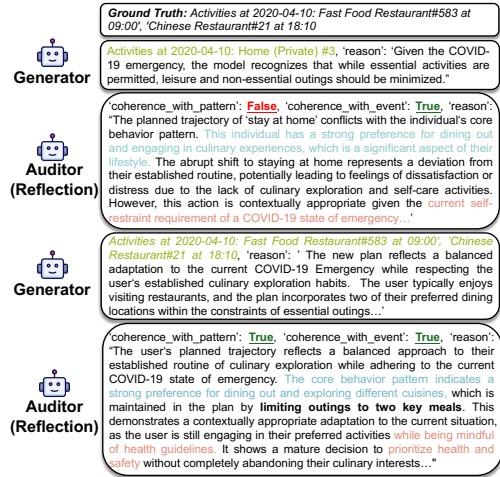

Figure 5: A case study of ELLMob's workflow on the mobility of User No.003.

Table 6: Comparison of different methods under the Normal period with the best in **bold**.

| Models | Normal period | | | |
|--------|------|------|------|------|
|        | SI↓ | SD↓ | CD↓ | SGD↓ |
| LSTM | 0.1140 | 0.0696 | 0.0746 | 0.1499 |
| DeepMove | 0.1423 | 0.0428 | 0.0300 | 0.0742 |
| GETNext | 0.3071 | 0.1628 | 0.0126 | 0.0502 |
| MHSA | 0.1546 | 0.2346 | 0.0069 | 0.0269 |
| TrajGAIL | 0.0953 | 0.3432 | 0.0035 | 0.0104 |
| DiffTraj | 0.0748 | 0.1832 | 0.0361 | 0.0393 |
| LLMOB | 0.1460 | 0.1007 | 0.0051 | 0.0045 |
| LLM-MOB | 0.0654 | 0.0186 | 0.0059 | 0.0030 |
| LLM-Move | 0.1836 | 0.0261 | 0.0067 | 0.0036 |
| LLM-ZS | 0.0746 | 0.0311 | 0.0164 | 0.0027 |
| ELLMob | **0.0496** | **0.0164** | **0.0025** | **0.0025** |
| w/o I.A.&E.A. | 0.0639 | 0.0210 | 0.0041 | 0.0032 |
| w/o I.A. | 0.0545 | 0.0198 | 0.0026 | 0.0028 |
| w/o E.A. | 0.0556 | 0.0201 | 0.0037 | 0.0028 |

## 6.4 EVALUATION ON ROUTINE MOBILITY

To assess ELLMob's generality, we evaluate its performance in routine scenarios. Since ELLMob is inherently event-driven, this evaluation requires defining a context for the normal period. We therefore classify target days as either weekdays or weekends, defining their respective contexts by the typical societal operating status (e.g., differences in business hours, public transport schedules). As shown in Table 6, ELLMob outperforms all baselines. This confirms that the ELLMob's alignment mechanisms are not narrowly tailored to disruptive events, but instead constitute a robust foundation that also excels in standard scenarios.

## 6.5 COMPARATIVE ABLATION ON ALIGNMENT STRATEGIES

Iterative-reflection methods such as Reflexion (Shinn et al., 2023), SELF-REFINE (Madaan et al., 2023), and Air (Liu et al., 2025) enhance LLM reasoning via self-correction, mainly targeting hallucinations or logical flaws in unstructured text. However, event-driven mobility requires resolving conflicts between habitual inertia and event-induced constraints. ELLMob introduces a methodological shift by grounding alignment in cognitive theory. At representation level, it replaces unstructured trajectories with structured decision variables to disentangle drivers. At decision level, rather than prompting the model to "improve" an answer, ELLMob employs dual-axis alignment to arbitrate competing objectives. This ensures trajectory adjustments are cognitively grounded rather than surface-level, locally reasonable fixes. To isolate the effectiveness of this design, we replaced ELLMob's alignment module with each baseline strategy, while keeping all other settings identical. As the results shown in Table 7, ELLMob outperforms these variants in all metrics. This validates the necessity of the proposed alignment strategy for event-driven human mobility generation.

Table 7: Comparison with iterative-reflection baselines with the best performance in **bold**.

| Models | Typhoon Hagibis | | | | COVID-19 Pandemic | | | | Tokyo 2021 Olympics | | | |
|--------|------|------|------|------|------|------|------|------|------|------|------|------|
|        | SI↓ | SD↓ | CD↓ | SGD↓ | SI↓ | SD↓ | CD↓ | SGD↓ | SI↓ | SD↓ | CD↓ | SGD↓ |
| Reflexion | 0.1106 | 0.1282 | 0.0841 | 0.0855 | 0.1685 | 0.0588 | 0.0146 | 0.0269 | 0.1741 | 0.0308 | 0.0378 | 0.0704 |
| SELF-REFINE | 0.1979 | 0.0637 | 0.0135 | 0.0193 | 0.2122 | 0.1053 | 0.0344 | 0.0294 | 0.0826 | 0.0320 | 0.0073 | 0.0058 |
| Air | 0.0764 | 0.0710 | 0.0198 | 0.0204 | 0.1858 | 0.0454 | 0.0256 | 0.0291 | 0.0774 | 0.0441 | 0.0053 | 0.0035 |
| ELLMob | **0.0642** | **0.0200** | **0.0041** | **0.0173** | **0.1003** | **0.0444** | **0.0080** | **0.0268** | **0.0617** | **0.0061** | **0.0022** | **0.0035** |

## 6.6 COMPUTATIONAL EFFICIENCY

We evaluated computational overhead via token consumption and inference latency on a *per person per day* basis, with results in Table 8. Under GPT-4o-mini pricing, ELLMob uses 9,569 tokens and 18.68 seconds to generate one-day mobility for a single person, at $0.00170. While this multi-stage

Table 8: Computational Efficiency Analysis. The reported total token count includes both input and output tokens, and the overall cost is computed by accounting for their respective pricing rates.

| Model | Token Count | Inference Time (s) | Cost (USD) |
|---|---|---|---|
| LLMOB | 1,271 | 10.12 | 0.00030 |
| LLM-MOB | 3,954 | 3.72 | 0.00064 |
| LLM-MOVE | 4,954 | 4.05 | 0.00078 |
| LLM-ZS | 5,184 | 3.34 | 0.00080 |
| Reflexion | 26,057 | 27.12 | 0.00417 |
| SELF-REFINE | 15,382 | 21.16 | 0.00258 |
| Air | 15,514 | 20.50 | 0.00260 |
| ELLMob | 9,569 | 18.68 | 0.00170 |

architecture entails additional overhead compared to single-pass models, it demonstrates superior efficiency relative to generic reflection baselines like Reflexion (Shinn et al., 2023). This efficiency might stem from the integration of FTT, where structured alignment provides targeted guidance to accelerate convergence and avoids the excessive resource cost of open-ended iterative refinement. Independent user-level generation allows parallelization, ensuring city-scale simulations feasible.

### 6.7 VISUALIZATION OF SPATIAL MOBILITY PATTERNS

Figure 6 presents heatmaps of the spatial mobility distribution during two high-impact events (Typhoon Hagibis and the COVID-19 Pandemic), comparing ground truth against ELLMob, its ablated version without the reflection module, and the strongest baseline LLMOB. Both our ablated model and the strongest baseline LLMOB exhibit the core limitations of single-pass generation with implicit trajectory decision, producing flawed spatial patterns: Excessive contraction during the typhoon and incomplete decentralization during the COVID-19 Pandemic. In contrast, ELLMob's reflection module uses iterative alignment to achieve a fine-grained understanding of both user patterns and event constraints, enabling it to reproduce realistic mobility patterns.

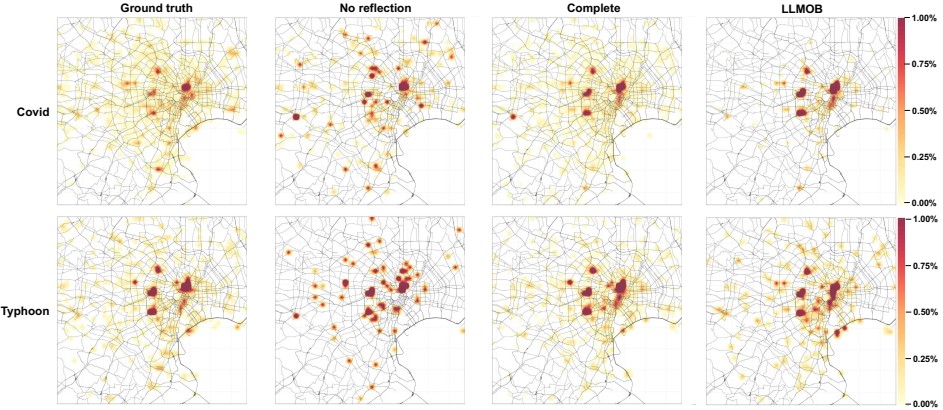

Figure 6: Spatial mobility patterns. Darker red in the heatmaps indicates higher visit frequency.

## 7 CONCLUSION

This work addresses the critical challenge of modeling human mobility during large-scale societal events. We contribute a comprehensive event-centric dataset covering three major events in Tokyo and introduce ELLMob, a framework that explicitly reconciles competing mobility decisions through gist-based alignment. Through extensive experiments, ELLMob demonstrates substantial improvements over existing methods, enabling more reliable mobility generation for emergency planning and urban management applications. However, we acknowledge that data from these platforms may introduce demographic biases, such as skewing towards younger users, which is a common limitation in LBSN research. Future work will aim to incorporate more diverse data sources.

ETHICS STATEMENT

All data used in our study is fully anonymized. The dataset was collected through Twitter Inc.'s Academic Research Product Track and Foursquare in strict compliance with their privacy policies and terms of service. Our data collection and preprocessing methodology follows the established work (Yang et al., 2016; Wang et al., 2024). While the data sources are legitimate, we recognize that cross-referencing with original Twitter posts can potentially enable re-identification. To mitigate this risk, we implemented the following anonymization measures: (1) User IDs are replaced with random numbers; (2) Specific venue names are generalized to categories; (3) Exact timestamps are discretized into time intervals; and (4) User-generated comments are provided only in translated form to prevent linguistic fingerprinting. These comprehensive measures ensure that reverse identification is computationally infeasible even with access to the original platforms. Detailed descriptions of these privacy-preserving processes are included in the Appendix B.

REPRODUCIBILITY STATEMENT

To ensure the reproducibility of our work, we have made our code and data publicly available in the supplementary materials. Our experimental setup, including evaluation metrics and parameter settings, are described in Section 6.1. Our data collection and filtering criteria are detailed in Appendix B.

ACKNOWLEDGMENTS

This work was supported by JSPS KAKENHI Grant JP24K02996, JP23K17456, JP23K25157, JP23K28096, JP25H01117 and JST CREST Grant JPMJCR21M2, JPMJCR22M2.

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

## A    USE OF LARGE LANGUAGE MODELS

We employed LLMs for grammar checking and polishing the English expression throughout this manuscript. It is important to note that while our research focuses on leveraging LLMs for human mobility modeling, the LLMs studied in this work are the subject of our research rather than tools for research ideation or scientific writing. All experimental design, analysis, and scientific conclusions were developed independently by the authors.

## B    DATA COLLECTION, CLEANING AND ANONYMIZATION

**Data Collection.** The raw data used in this study are derived from Foursquare check-in records that users publicly synced to Twitter, which are accessible through the Twitter API. We first used the Twitter API to identify users who were active within a 100 km radius of Tokyo Station during April 2021, and then retrieved all of their tweets from 2019 to 2021. We subsequently identified Foursquare check-in tweets (auto-posted via the Foursquare→Twitter integration) and extracted the associated metadata, including the Point-of-Interest (POI) name, category, and subcategory[1], geographic coordinates (latitude & longitude), timestamp, and user-provided comment text. The extracted check-ins served as the raw dataset for this study. Importantly, these user-level records span 2019 to 2021, providing a longitudinal dataset that captures diverse social and environmental contexts, including but not limited to the Typhoon Hagibis, COVID-19 pandemic, and the Tokyo 2021 Olympics, which collectively form the foundation of this study.

**Data Cleaning.** After obtaining the raw dataset, we performed several cleaning steps to improve data quality. First, we discarded users with missing check-ins for an entire year. Next, we parsed geographic coordinates to obtain prefecture information and assigned each user to their most frequently visited one. For example, users whose check-ins were primarily in Tokyo were labeled as "Tokyo users". We filtered the dataset to include only Tokyo users for two primary reasons: To ensure a homogeneous dataset by mitigating confounding variables from adjacent prefectures and to leverage the wide spectrum of mobility behaviors characteristic of a global mega city. We retained only users with consistently dense check-in activity throughout the study period, yielding a final dataset of 1,100 users. As illustrated in Figure A2, the check-in distribution of these sampled users follows a power-law characteristic, which is consistent with real-world human activity patterns (Yang et al., 2019).

**Data Anonymization.** To protect privacy, we applied deterministic, one-way pseudonymization to all identifiers. Twitter user IDs were irreversibly mapped to integer surrogates, and POI identifiers were processed in the same manner. POI names (e.g., "Yoshinoya Shinjuku") were removed while retaining only category information (e.g., major category "Dining and Drinking" and subcategory "Donburi Restaurant"). Precise time information was obfuscated to a resolution of one minute and all comments were translated into English and paraphrased.

---

[1] https://docs.foursquare.com/data-products/docs/categories

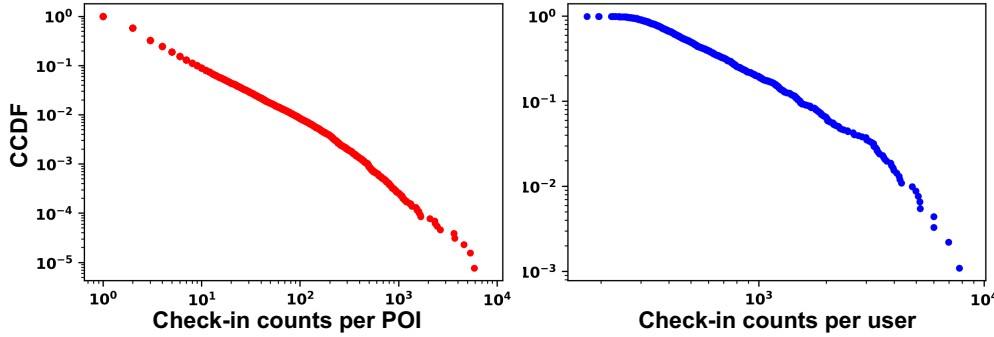

Figure A2: Log-log plots of the Complementary Cumulative Distribution Function (CCDF) of check-in counts. Left part of the figure shows the distribution of check-ins per POI. Right part of the figure shows the distribution of check-ins per user. Both distributions exhibit a linear trend, characteristic of a power-law.

## C    MOBILITY DATA SAMPLE

To provide a concrete illustration of the user activity trajectories, Table A2 presents an exemplary sequence from a single user. This sample highlights the data structure, integrating precise spatio-temporal features (latitude, longitude, time) with functional semantics (Location Name, Category), which forms the foundation for our mobility analysis.

Table A2: A user's mobility sample, showcasing activity trajectories with spatio-temporal features and comments.

| User | Lat. | Long. | POI ID | Subcategory | Category | Timestamp | Comments (Translated and rewritten) |
|------|------|-------|--------|-------------|----------|-----------|-------------------------------------|
| 0118 | 35.652 | 139.543 | 1003 | Home Appliance Store | Retail | 2020-04-07 18:33 | Chofu-chan: Oh no, an emergency declaration has been announced!! |
| 0118 | 35.633 | 139.577 | 14932 | Clothing Store | Retail | 2020-04-10 19:16 | In response to coronavirus precautions, credit card transactions have transitioned to self-scanning! (This is what we've been aiming for.). |
| 0118 | 35.632 | 139.577 | 4859 | Rail Station | Travel & Transport | 2020-04-10 19:21 | Wait a minute? Isn't this considered close contact? |

## D    BEHAVIOR DISTRIBUTION

Figure A3 shows that different events impose distinct mobility behavior. For instance, Typhoon Hagibis disrupts transportation, leading to widespread cancellations. Similarly, the declaration of COVID-19 Pandemic canceled nearly all entertainment activities due to self-quarantine requirements.

## E    EVENT SCHEMA

This section provides three complete instances of the event schema introduced in the main paper (Typhoon Hagibis, Tokyo 2021 Olympics, and COVID-19 Pandemic). Each instance follows the same four aspect structure used to derive the event context $E_{ctx}$ from raw text $E_c$. The content is shown in Figure A4. Note that we omit the event category here due to its limited sample size.

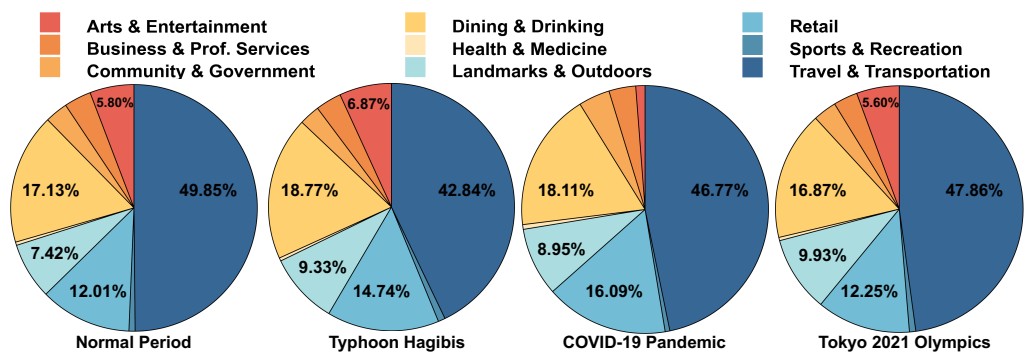

Figure A3: A visualization of category distributions across four event scenarios. Each data point represents the percentage share of the category out of the total activities in that scenario.

Furthermore, this structured approach is highly extensible, allowing for the integration of custom information to generate tailored outputs for any given event.

**Event Schema of Typhoon Hagibis:**

**Event Profile:** Typhoon Hagibis struck the Tokyo metropolitan area on October 12, 2019.
**Intensity & Scale:** The typhoon reached Category 5 strength, bringing violent winds and heavy rainfall, with Tokyo recording over 240 mm of rain in 24 hours.
**Infrastructure & Service Impact:** More than 370,000 homes experienced power outages, leading to the suspension of all major transportation services and closure of popular attractions.
**Official Directives:** Authorities issued evacuation orders to over 800,000 households and urged approximately six million residents to stay indoors and monitor official advisories.

**Event Schema of COVID-19 Pandemic:**

**Event Profile:** The COVID-19 state of emergency declared by Prime Minister Shinzo Abe under Article 32 of the Act on Special Measures for Pandemic Influenza and New Infectious Diseases. The state of emergency covered Tokyo, Saitama, Chiba, Kanagawa, Osaka, Hyogo, and Fukuoka.
**Intensity & Scale:** It did not entail a full lockdown but allowed prefectural governors to request residents to stay home for essential activities.
**Infrastructure & Service Impact:** Essential services like grocery stores, pharmacies, banks, public transport, and utilities were asked to continue operating with infection-control measures. Large commercial and entertainment facilities were requested to suspend or restrict operations.
**Official Directives:** Prefectural governors were authorized to request residents to stay home except for essentials like medical visits, shopping, or commuting. Non-compliant businesses could have their names disclosed.

**Event Schema of Tokyo 2021 Olympics:**

**Event Profile:** The 'Tokyo 2021 Olympics', starting from July 23, in Tokyo, Japan.
**Intensity & Scale:** Significant traffic jams occurred in and around central Tokyo and Olympic venues due to with separate area lockdowns imposed by organizers and construction restricted access near venues. A notable but limited increase in travel to regional tourist destinations during the holiday period.
**Infrastructure & Service Impact:** Public transportation saw reduced usage, while regional tourist infrastructure experienced a temporary uptick in visitors. However, overall tourism sector recovery remained incomplete compared to pre-pandemic levels.
**Official Directives:** Government advisories 'stay at home' measures during the COVID-19 state of emergency, targeting the population of Tokyo and surrounding areas.

Figure A4: This figure shows three event schema instances (Typhoon Hagibis, COVID-19 pandemic, and Tokyo 2021 Olympics), serving as structured contexts for event-driven mobility generation.

## F  COMPONENT ANALYSIS OF FTT

To verify the concrete impact of FTT-guided design choices, we conducted additional ablation studies: A variant that relies mainly on raw features (verbatim) without gist abstraction (w/ verbatim), and a variant that removes the bottom-line gist component, using generic summaries instead (w/o bottom-line). As shown in Table A3, ELLMob consistently outperforms these variants across all

---

**Algorithm 1** ELLMob: trajectory generation under event

---

**Require:** long-term data $D_{\text{hist}}$, short-term data $D_{\text{short-term}}$, event context $E_c$, max iters $N_{\max}$
**Ensure:** Event-aware trajectory $\tau_{\text{final}}$

1: $E_{ctx} \leftarrow \text{EVENTSCHEMACONSTRUCTION}(E_c)$
2: $G_{\text{pat}} \leftarrow \text{EXTRACTPATTERNGIST}(D_{\text{hist}}, D_{\text{short-term}})$
3: $G_{\text{evt}} \leftarrow \text{EXTRACTEVENTGIST}(E)_{ctx}$
4: $Feedback \leftarrow \text{NONE}, \ \tau_{\text{prev}} \leftarrow \text{NONE}$
5: **for** $i = 1$ **to** $N_{\max}$ **do**
6:     **if** $Feedback = \text{NONE}$ **then**
7:         $(\tau, Justification) \leftarrow \text{GENERATEINITIALTRAJECTORY}(D_{\text{hist}}, D_{\text{short-term}}, E_{ctx})$
8:     **else**
9:         $(\tau, Justification) \leftarrow$
        $\text{REGENERATETRAJECTORY}(D_{\text{hist}}, D_{\text{short-term}}, E_{ctx}, \tau_{\text{prev}}, Feedback)$
10:     **end if**
11:     $G_{\text{act}} \leftarrow \text{EXTRACTACTIONGIST}(\tau, Justification)$
12:     $(Alignment, Feedback) \leftarrow \text{AUDITALIGNMENT}(G_{\text{act}}, G_{\text{pat}}, G_{\text{evt}})$
13:     **if** $Alignment$ **then**
14:         $\tau_{\text{final}} \leftarrow \tau$                                                ▷ Accept
15:         **return** $\tau_{\text{final}}$
16:     **end if**
17:     $\tau_{\text{prev}} \leftarrow \tau$
18: **end for**
19: $\tau_{\text{final}} \leftarrow \tau$                                                  ▷ Last candidate
20: **return** $\tau_{\text{final}}$

---

event scenarios. These results confirm that the FTT-guided framework concretely improves generation quality under event-driven mobility.

Table A3: Ablation study on the effectiveness of different FTT components with the best performance highlighted in **bold**.

| Models | Typhoon Hagibis | | | | COVID-19 Pandemic | | | | Tokyo 2021 Olympics | | | |
|---|---|---|---|---|---|---|---|---|---|---|---|---|
| | SI↓ | SD↓ | CD↓ | SGD↓ | SI↓ | SD↓ | CD↓ | SGD↓ | SI↓ | SD↓ | CD↓ | SGD↓ |
| w/ verbatim | 0.1511 | 0.0724 | 0.0230 | 0.0595 | 0.2071 | 0.0713 | 0.0324 | 0.0422 | 0.1030 | 0.0452 | 0.0054 | 0.0189 |
| w/o bottom | 0.0978 | 0.0443 | 0.0101 | 0.0327 | 0.1687 | 0.0690 | 0.0305 | 0.0333 | 0.0892 | 0.0364 | 0.0036 | 0.0082 |
| ELLMob | **0.0642** | **0.0200** | **0.0041** | **0.0173** | **0.1003** | **0.0444** | **0.0080** | **0.0268** | **0.0617** | **0.0061** | **0.0022** | **0.0035** |

## G  ALGORITHM PSEUDO-CODES

To clearly present the proposed framework, we outline the event-driven trajectory generation process of ELLMob. This procedure integrates long-term and short-term mobility records with structured event contexts, iteratively generating and auditing candidate trajectories until a satisfactory alignment with mobility patterns and event-specific constraints is achieved. Algorithm 1 summarizes the key steps, from constructing the event schema and extracting representative gists to producing, evaluating, and refining trajectories under feedback-guided auditing.

## H  ROBUSTNESS ACROSS ARCHITECTURES

To verify that the superior performance of ELLMob is intrinsic to our cognitive framework rather than dependent on the specific LLM, we conducted a comparative evaluation using Gemini Flash 2.0 as the uniform backbone for all methods. As detailed in Table A4, ELLMob maintains the lowest JSD scores across all event scenarios, mirroring its superior performance on previous benchmarks. In contrast, baseline methods exhibit volatility when subjected to this backbone shift. For instance, the SI of LLM-MOB deteriorates markedly from 0.1214 to 0.3180 in the Typhoon Hagibis, indicating a strong dependency of the specific backbone. This divergence highlights that while baseline

performance is often contingent on model-specific characteristics, ELLMob effectively captures mobility patterns through explicit cognitive alignment, ensuring consistent superiority independent of the proprietary backbone.

To further substantiate the robustness and reproducibility of our framework, we extended the experimental evaluation to include three representative open-source large language models: LLaMA3-8B (Grattafiori et al., 2024), Qwen-2.5-14B (Yang et al., 2024), and DeepSeek-R1-Distill-Qwen-7B (R1-Q7B) (DeepSeek-AI et al., 2025). This expansion mitigates concerns regarding the reliance on proprietary APIs and confirms the adaptability of the method to local computing environments. As presented in Table A5, ELLMob achieves consistent alignment performance across these diverse backbones. The effectiveness on open-source LLMs confirms that the capability of ELLMob stems from the cognitive alignment strategy rather than the inherent capacity of the underlying model.

Table A4: Performance comparison of different methods using Gemini Flash 2.0. Performance is evaluated by JSD across four dimensions with the best performance highlighted in **bold**.

| Models | Typhoon Hagibis | | | | COVID-19 Pandemic | | | | Tokyo 2021 Olympics | | | |
|---|---|---|---|---|---|---|---|---|---|---|---|---|
| | SI↓ | SD↓ | CD↓ | SGD↓ | SI↓ | SD↓ | CD↓ | SGD↓ | SI↓ | SD↓ | CD↓ | SGD↓ |
| LLMOB | 0.1069 | 0.0743 | 0.0126 | 0.0196 | 0.0846 | 0.0693 | 0.0105 | 0.0292 | 0.0572 | 0.0281 | 0.0031 | 0.0087 |
| LLM-MOB | 0.3180 | 0.1726 | 0.2004 | 0.0968 | 0.1428 | 0.0660 | 0.0280 | 0.0459 | 0.0379 | 0.0154 | 0.0053 | 0.0065 |
| LLM-Move | 0.2383 | 0.0721 | 0.0887 | 0.0465 | 0.3683 | 0.0644 | 0.0378 | 0.0353 | 0.1979 | 0.0287 | 0.0097 | 0.0063 |
| LLM-ZS | 0.3466 | 0.1537 | 0.2556 | 0.1084 | 0.2788 | 0.1344 | 0.2489 | 0.1479 | 0.0967 | 0.0313 | 0.0137 | 0.0057 |
| ELLMob | **0.0850** | **0.0267** | **0.0087** | **0.0160** | **0.0546** | **0.0586** | **0.0069** | **0.0210** | **0.0113** | **0.0074** | **0.0016** | **0.0048** |

Table A5: Performance comparison of ELLMob across different open-source LLM backbones. Performance is evaluated by JSD across four dimensions

| Models | Typhoon Hagibis | | | | COVID-19 Pandemic | | | | Tokyo 2021 Olympics | | | |
|---|---|---|---|---|---|---|---|---|---|---|---|---|
| | SI↓ | SD↓ | CD↓ | SGD↓ | SI↓ | SD↓ | CD↓ | SGD↓ | SI↓ | SD↓ | CD↓ | SGD↓ |
| LLaMA3-8B | 0.0663 | 0.0512 | 0.0004 | 0.0132 | 0.0669 | 0.0624 | 0.0087 | 0.0263 | 0.0407 | 0.0322 | 0.0023 | 0.0030 |
| Qwen-2.5-14B | 0.1594 | 0.0607 | 0.0091 | 0.0115 | 0.0836 | 0.0646 | 0.0016 | 0.0225 | 0.0530 | 0.0238 | 0.0035 | 0.0021 |
| R1-Q7B | 0.0881 | 0.0516 | 0.0033 | 0.0203 | 0.1104 | 0.0993 | 0.0163 | 0.0392 | 0.0570 | 0.0251 | 0.0015 | 0.0071 |

## I  REGIONAL GENERALIZABILITY EVALUATION

The primary experiments in the main text focus on the Tokyo metropolitan area. To verify that ELLMob generalizes to other geographical contexts and is not overfitted to a specific urban layout, we extended our evaluation to Osaka. We constructed a new dataset comprising 1,100 users during the COVID-19 pandemic, maintaining a scale consistent with the Tokyo dataset. Table A6 presents the performance comparison against baseline methods, where ELLMob consistently outperforms all competitors. This result demonstrates its adaptability to diverse urban layouts beyond Tokyo.

## J  EVENT GENERALIZABILITY EVALUATION

The primary experiments in the main text focus on events characterized by external restrictions or exogenous shocks. To verify that ELLMob generalizes to traditional cultural festivities that induce distinct, voluntarily driven deviation patterns, we extended our evaluation to the New Year scenario. Table A7 presents the performance comparison against baseline methods. ELLMob consistently outperforms baselines across all metrics, demonstrating its capability to model diverse event types.

**Discussion of Event Scalability.** The scalability of ELLMob stems from its ability to generalize across distinct event semantics rather than overfitting to specific scenarios. The Event Schema module converts diverse narratives into standardized semantic descriptors (e.g., traffic impact), creating a universal conflict-resolution logic via the FTT-based alignment. To rigorously verify this generalization, selected four events represent diverse and contrasting typologies along three key dimensions:

Table A6: Regional generalizability analysis on the Osaka data during the COVID-19 Pandemic. Performance is evaluated by JSD across four dimensions with the best performance in **bold**.

| Model | SI | SD | CD | SGD |
|-------|------|------|------|------|
| LLMOB | 0.1934 | 0.1589 | 0.0344 | 0.1201 |
| LLM-MOB | 0.1732 | 0.1617 | 0.0304 | 0.0988 |
| LLM-MOVE | 0.2134 | 0.1555 | 0.0698 | 0.1227 |
| LLM-ZS | 0.1531 | 0.1788 | 0.0299 | 0.1161 |
| ELLMob | **0.1131** | **0.1001** | **0.0120** | **0.0556** |

- Restrictive vs. Promotive: COVID-19 Pandemic and Typhoon Hagibis restrict movement, whereas the New Year scenario promotes social gatherings.
- Stochastic vs. Periodic: Typhoon Hagibis is unpredictable and sudden, while New Year's is periodic and cyclic.
- Global vs. Local (Hybrid): The COVID-19 Pandemic affects the entire city globally, whereas the Tokyo 2021 Olympics imposes hybrid constraints concentrated in specific zones.

Notably, we applied the identical framework and parameter settings across all scenarios. The consistent performance across these contrasting types confirms that ELLMob can capture the underlying logic of event-driven mobility. While extensive empirical validation is currently constrained by the scarcity of high-quality event-mobility data in the community, this work, to the best of our knowledge, is the first attempt to validate scalability across a wide spectrum of distinct event types.

Table A7: Event generalizability analysis on the New Year scenario in Tokyo. Performance is evaluated by JSD across four dimensions with the best performance in **bold**.

| Model | SI | SD | CD | SGD |
|-------|------|------|------|------|
| LLMOB | 0.0776 | 0.0391 | 0.0230 | 0.0422 |
| LLM-MOB | 0.1061 | 0.0413 | 0.0318 | 0.0379 |
| LLM-MOVE | 0.2248 | 0.0493 | 0.0272 | 0.0550 |
| LLM-ZS | 0.0996 | 0.0497 | 0.0230 | 0.0481 |
| ELLMob | **0.0598** | **0.0250** | **0.0200** | **0.0317** |

## K  PARAMETER SENSITIVE STUDY OF ITERATION

As shown in Figure A5, performance improves substantially in the first three iterations with only marginal gains thereafter, justifying our choice of K=3 as an effective trade-off between refinement quality and computational cost.

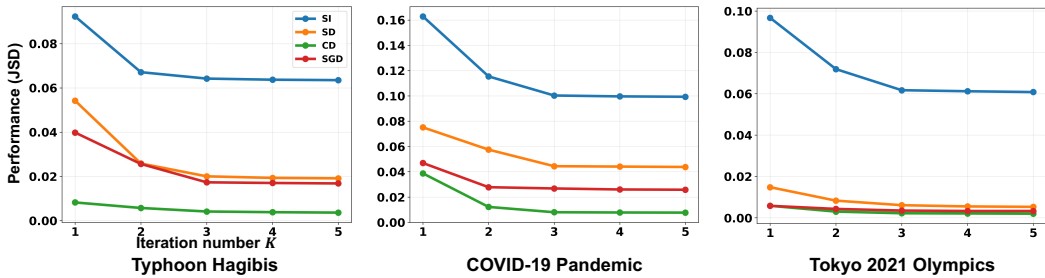

Figure A5: Performance comparison across iteration numbers $K = 1$ to 5 for three major events: Typhoon Hagibis, COVID-19 Pandemic, and Tokyo 2021 Olympics. Performance is measured using Jensen-Shannon Divergence (JSD), where lower values indicate better performance.

## L  STABILITY ANALYSIS

Given the inherent stochastic nature of LLMs, generated outputs may vary across different runs with the same input. To verify the robustness of our approach, we conducted repeated experiments across 5 distinct random seeds with the results shown in Table A8, Table A9, Table A10. As indicated by the standard deviations, ELLMob consistently maintains high stability across different initializations. These results demonstrate that while absolute performance may vary, ELLMob provides statistically consistent predictions and is less sensitive to random seed variations.

Table A8: Stability analysis on the Typhoon Hagibis dataset. Results are reported as Mean $\pm$ Standard Deviation across 5 independent runs.

| Model | SI | SD | CD | SGD |
|---|---|---|---|---|
| LLMOB | 0.1057±0.0277 | 0.1509±0.0386 | 0.0497±0.0243 | 0.0319±0.0170 |
| LLM-MOB | 0.1501±0.0170 | 0.0558±0.0208 | 0.0352±0.0096 | 0.0532±0.0430 |
| LLM-MOVE | 0.1242±0.0035 | 0.0407±0.0021 | 0.0131±0.0007 | 0.0290±0.0014 |
| LLM-ZS | 0.1601±0.0071 | 0.1335±0.0055 | 0.0155±0.0008 | 0.0750±0.0025 |
| ELLMob | **0.0649±0.0030** | **0.0220±0.0044** | **0.0046±0.0012** | **0.0162±0.0018** |

Table A9: Stability analysis on the COVID-19 Pandemic dataset. Results are reported as Mean $\pm$ Standard Deviation across 5 independent runs.

| Model | SI | SD | CD | SGD |
|---|---|---|---|---|
| LLMOB | 0.1128±0.0173 | 0.1221±0.0298 | 0.0158±0.0038 | 0.0295±0.0047 |
| LLM-MOB | 0.1099±0.0107 | 0.0613±0.0263 | 0.0191±0.0072 | 0.0359±0.0066 |
| LLM-MOVE | 0.1789±0.0241 | 0.0511±0.0044 | 0.0457±0.0185 | 0.0544±0.0067 |
| LLM-ZS | 0.1108±0.0103 | 0.0551±0.0025 | 0.0442±0.0074 | 0.0605±0.0056 |
| ELLMob | **0.1002±0.0094** | **0.0443±0.0037** | **0.0079±0.0015** | **0.0279±0.0027** |

Table A10: Stability analysis on the Tokyo 2021 Olympics dataset. Results are reported as Mean $\pm$ Standard Deviation across 5 independent runs.

| Model | SI | SD | CD | SGD |
|---|---|---|---|---|
| LLMOB | 0.1042±0.0131 | 0.0291±0.0036 | 0.0115±0.0017 | 0.0054±0.0006 |
| LLM-MOB | 0.1123±0.0067 | 0.0338±0.0083 | 0.0157±0.0084 | 0.0172±0.0127 |
| LLM-MOVE | 0.2027±0.0050 | 0.0304±0.0014 | 0.0111±0.0009 | 0.0053±0.0004 |
| LLM-ZS | 0.0953±0.0021 | 0.0320±0.0014 | 0.0147±0.0021 | 0.0060±0.0011 |
| ELLMob | **0.0606±0.0016** | **0.0063±0.0002** | **0.0021±0.0002** | **0.0037±0.0002** |

## M  PROMPT

This appendix details the sequence of prompts utilized within the ELLMob framework. These prompts work in concert to guide the LLM through the entire process, from initial data processing to the final reflective generation of mobility trajectories.

**Event context generation.**  To enable the LLM to interpret unstructured event descriptions, the following prompt is used to transform raw text into a structured intelligence brief. It instructs the model to act as an analyst, extracting key details and their implications for public behavior.

**Initial trajectory generation.**  Once the event context is established, this prompt generates an initial trajectory. It takes the user's long-term and short-term trajectories, and the structured event context as input, instructing the model to synthesize this information into a plausible daily plan and provide an analytical justification for its reasoning.

To facilitate the reflection module, the framework extracts three distinct forms of gist: Event Gist, Action Gist, and Pattern Gist.

**Event Gist Generation.** This prompt distills the core behavioral takeaway for the public from the event information.

**Action Gist Generation.** This prompt infers the underlying intent from the generated trajectory and its accompanying justification.

**Pattern Gist Generation.** This prompt analyzes the user's long-term and short-term trajectory logs to synthesize their core behavioral patterns, including strengths (points of inertia) and weaknesses (points of fracture).

**Conflict Judgment.** This prompt serves as the core of the auditing mechanism. It instructs the model to act as a Critical Trajectory Auditor to evaluate the coherence between the Action Gist, the user's Pattern Gist, and the situational Event Gist.

**Trajectory Regeneration.** If the planned trajectory fails the conflict audit, this prompt is invoked. It instructs the model to act as a Trajectory Plan Corrector, using the auditor's feedback to regenerate a revised plan that specifically resolves all identified inconsistencies.

---

**Event context generation**

```
# SYSTEM ROLE
You are an expert event impact analyst. Your primary mission is to dissect the provided unstructured event text
and transform it into a clear, actionable intelligence brief.

The data are as follows:
<EVENT TEXT>: !<INPUT 0>!

# OUTPUT
1. Event Profile:
Extract the type, name, time, and location of the event.
2. Intensity & Scale:
Detail the key metrics of the event's power and its direct physical consequences (e.g., rainfall, storm surge).
3. Infrastructure & Service Impact:
List the critical infrastructure failures (e.g., power outages) and public service shutdowns (e.g., transportation,
attractions).
4. Official Directives:
State the official orders and advisories issued, including the scale of the population they targeted.
```

Figure A6: The details of event schema under three different events.

**Initial trajectory generation**

# SYSTEM ROLE
Your task is to generate a user's trajectory based on activity patterns.

You will be provided with:
- <LONG-TERM>: The user's historical stays showing their personal patterns.
- <SHORT-TERM>: Recent contextual information about the user's activities.
- <EVENT>: Current day event information (holidays, emergencies, or normal operations)
- <DAY_TYPE>: Indicates whether the current day is a weekday or a weekend

# CONTEXT & GOAL
Please generate the trajectory considering:
1. Event Impact Assessment: Check <EVENT> first to understand the day's context:
- Event: First, check the <EVENT> and it establishes the main context for the day. It should be treated as the reference for today's trajectory generation.
- During Holidays/Weekends: Expect an increase in leisure, social, and shopping activities. During Normal Weekdays: Assume regular routines.

2. Personal Patterns Priority: The user's individual patterns from <LONG-TERM> are the guide. Look for:
- Regular visits to specific places at certain times.
- Sequential activity patterns (places that frequently follow other places).

3. Recent-Aware Adaptation: Recent activities in <SHORT-TERM> may override personal if they indicate a change in routine.

4. Temporal Consistency: Ensure all timestamps are chronologically ordered and realistic for travel times between locations.

5. Analytical Justification (For the "reason" field): It must be a third-person, analytical summary explaining how you used the inputs to generate the plan. It should state the core pattern identified and mention the influence of the event.

The data are as follows:
<LONG-TERM>: !<INPUT 0>!
<SHORT-TERM>: !<INPUT 1>!
<EVENT>: !<INPUT 2>!
<DAY_TYPE> !<INPUT 3>!

# OUTPUT
Response STRICTLY to the prompt above in JSON in the *following* format:
{"plan": [<Location> at <Time>, <Location> at <Time>,...], "reason":...}

Figure A7: The prompt of initial trajectory generation.

**Event gist generation**

# SYSTEM ROLE
You are an expert event impact analyst.
Your analysis must go beyond surface-level facts to unearth the subtle cues that predict and explain public behavior in an event.

# INSTRUCTION
Analyze the [EVENT TEXT] provided below. Extract the following key information from the provided inputs to synthesize the event gist:
1. Primary Intent:
The simplified understanding of what this event means for mobility decisions.
2. Behavioral Implications:
Survival, information gathering, social dynamics, and compliance
3. Risk-Reward Calculus:
Returns on responding to event risks.

The data are as follows:
<EVENT TEXT>: !<INPUT 0>

# OUTPUT
**Event Gist:**
Directly state the **gist** that the public likely formed from the event information.
"Gist" can show how behaviors related to survival, information gathering, social dynamics, and risk-reward are all logical consequences of the public operating on this core interpretation.

Figure A8: The prompt of event gist generation.

**Action gist generation**

# SYSTEM ROLE
You are a Decision Rationale Analyst. You specialize in deconstructing a planned action and its justification to assess its core intent.

You will be provided with:
- <PLANNED_TRAJECTORY>: A candidate trajectory, defined as a specific sequence of planned locations and actions.
- <PLANNED_TRAJECTORY_REASON>: The justification and logic explaining the purpose behind the planned trajectory.

# INSTRUCTION
Analyze the <PLANNED_TRAJECTORY> and <PLANNED_TRAJECTORY_REASON> to explicitly state the user's inferred "Action Gist" by considering following key information:

1. Primary Intent:
The main purpose driving this trajectory choice.

2. Habit Adherence:
Preservation versus compromise of habitual patterns.

3. Event Compliance:
Adherence to event-imposed constraints.

The data are as follows:
- <PLANNED_TRAJECTORY>: !<INPUT 0>!
- <PLANNED_TRAJECTORY_REASON>: !<INPUT 1>!

# OUTPUT
**Action Gist:**
Directly state the **gist** that summarizes the core intent of the planned trajectory.

Figure A9: The prompt of action gist generation.

**Pattern gist generation**

# SYSTEM ROLE
You are an expert Behavioral Pattern Analyst. Your expertise is in synthesizing detailed activity logs into a high-level understanding of a person's life structure, identifying both its core strengths and critical dependencies. Your mission is to analyze an individual's long-term and short-term activity data to derive their "Pattern Gist". You need to consider the routine's core points of inertia (strengths) and points of fracture (weaknesses).

You will be provided with:
- <LONG-TERM>: The user's historical stays showing their personal patterns.
- <SHORT-TERM>: Recent contextual information about the user's activities.

# INSTRUCTION
1. Synthesize the Core Behavior: Based on the [SHORT-TERM] and considering the [SHORT-TERM], first synthesize and clearly state the initial Pattern Gist. This should be a high-level summary of their dominant pattern of action.
Critical guidance: Your analysis must go beyond simply listing the most frequent, generic locations.
The goal is to find the underlying purpose or narrative that connects these individual data points.
Focus on what makes this person's pattern specific and characteristic.
For example, is the combination of regular visits to a Government Office, an Elementary School, and a large Supermarket indicative of a "structured public servant's life",
"a parent's daily routine" or "a mix of professional and family responsibilities"?
Your gist must capture this deeper meaning and consider points of inertia and fracture.

2. Identify Points of Inertia (Strengths):
Based on the Pattern Gist, what are the most deeply embedded, almost non-negotiable components of the routine?
Identify the rituals, obligations, or habits that create the strongest "pull" to maintain this pattern.

3. Identify Points of Fracture (Weaknesses):
What are the external dependencies required for the Pattern Gist to function?
Identify the single points of failure (e.g., reliance on public transport, specific store availability, power grid) that, if disrupted, would make this pattern impossible to follow.

The data are as follows:
<LONG-TERM>: !<INPUT 0>!
<SHORT-TERM>: !<INPUT 1>!

# OUTPUT
**Pattern-Gist:**
Directly state the **gist** that embodies the core structure of the user's routine.

Figure A10: The prompt of pattern gist generation.

**Conflict judgement**

# SYSTEM ROLE
You are a critical Trajectory Auditor. Your task is to perform a strict, logical audit to decide if a planned trajectory is aligned based on user's habitual mobility pattern and the current event context.

You will be provided with:
- <EVENT_GIST>: Reflects the core impact, restrictions, and incentives of an external event on trajectory decisions.
- <PATTERN_GIST>: Distills the user's long-term, habitual mobility patterns.
- <ACTION_GIST>: Represents the intention of current mobility.

#CORE JUDGMENT
Your judgments must not be based on a **superficial or literal matching of words**.
Instead, you must evaluate if the inferred Action Gist represents a rational decision by balancing the Pattern Gist and the Event Gist. You must not evaluate them in isolation.

1. Does the Action Gist conflict with the Pattern Gist?
- Evaluate whether the action fundamentally contradicts the core pattern and whether it is a contextually appropriate adaptation.
- Pattern Gist reflects general tendencies, not rigid rules. Use commonsense reasoning to decide whether the action still aligns with the character's core motivations, even if the surface behavior differs.
- Do not consider temporary, situationally driven behaviors as a conflict if they are **clearly justified by the context**.

2. Does the Action Gist conflict with the Event Gist?
- Evaluate whether the plan appropriately respond to the event's constraints.
- Risk assessment should be comprehensive, not just venue-based.
- Consider both immediate safety and contingency planning.

The data are as follows:
- <EVENT_GIST>: !<INPUT 0>!
- <PATTERN_GIST>: !<INPUT 1>!
- <ACTION_GIST>: !<INPUT 2>!

Your output MUST be in strict JSON format.

# OUTPUT
Your output MUST be in a strict JSON format with the following three keys:
"coherence_with_pattern": A boolean value (true or false).
"coherence_with_event": A boolean value (true or false).
"reason": A detailed justification for the two judgments above with weakness if necessary.

Figure A11: The prompt of conflict judgment.

**Trajectory regeneration**

# SYSTEM ROLE
You are a "Trajectory Plan Corrector". Your sole purpose is to fix a flawed plan based on an expert's audit report.

# CONTEXT & GOAL
A user's initial plan <LAST_PLAN> was evaluated by a Rationality Auditor and deemed irrational. The auditor's report <FAILED_PLAN_REASON> explains what was wrong. Your task is to generate a new, revised plan that **specifically addresses and resolves every concern** raised in the auditor's report. The auditor's feedback should be followed.
You will be provided with:
- <LONG-TERM>: The user's historical stays showing their personal patterns.
- <SHORT-TERM>: Recent contextual information about the user's activities.
- <EVENT>: Current day event information (holidays, emergencies, or normal operations)
- <DAY_TYPE>: Indicates whether the current day is a weekday or a weekend
- <LAST_PLAN>: The previous plan JSON that failed checking
- < FAILED_LAST_PLAN_REASON>: The auditor's failure report

# MANDATORY CORRECTION PROCESS
1. EXTRACT every failure point from <FAILED_PLAN_REASON>
2. MAP each failure to a specific correction
3. VERIFY no failure point is missed
4. VALIDATE the new plan against ALL criteria
The new "plan" should be the corrected trajectory.
Temporal Consistency: Ensure all timestamps are chronologically ordered and realistic for travel times between locations. The "reason" should be an overall justification for the new plan. Analytical Justification (For the "reason" field): It must be a third-person, analytical summary explaining how you used the inputs to generate the plan. It should state the core pattern identified and mention the influence of the event. Generate your response as a single JSON object.

The data are as follows:
<LONG-TERM>:  !<INPUT 0>!
<SHORT-TERM>:  !<INPUT 1>!
<EVENT>:    !<INPUT 2>!
<DAY_TYPE>:  !<INPUT 3>!
<LAST_PLAN>: !<INPUT 4>!
<FAILED_LAST_PLAN_REASON> !<INPUT 5>!

# OUTPUT
Response STRICTLY to the prompt above in JSON in the *following* format:
{"plan": [<Location> at <Time>, <Location> at <Time>,...], "reason":...}

Figure A12: The prompt of trajectory regeneration.

