# OpenReview forum: "ELLMob: Event-Driven Human Mobility Generation with Self-Aligned LLM Framework"
_ICLR.cc/2026/Conference — ICLR 2026 Poster_

### Official Review · Reviewer_k5vy · 2025-10-18

**Soundness:** 2
**Presentation:** 4
**Contribution:** 2
**Rating:** 4
**Confidence:** 3

**Summary:**

The paper proposes ELLMob, an event-driven human mobility generation framework that leverages large language models (LLMs) to synthesize realistic human trajectories under large-scale societal events. Unlike existing approaches that mainly learn habitual mobility patterns, ELLMob explicitly models the competition between routine behaviors and event-induced disruptions (e.g., natural disasters, pandemics, or major events). Following this, this paper proposed the first human-mobility dataset that is annotated with the event label, and then proposed an LLM-based method with ''reflect–refine'' paradigm for human mobility generation.

**Strengths:**

(1) The paper tackles event-driven mobility generation, a previously overlooked challenge. While most prior works model routine movements, this study explicitly considers non-routine deviations caused by major events—a critical step toward urban simulations, especially contexted by big social event.

(2) The paper is well-writen and the proposed method is technically sound.

(3) This paper contributes a new dataset that has event-label  along with the human trajectory data.

**Weaknesses:**

(1) One concern is number of events in the paper, which only covers Typhoon, COVID-19 Pandemic, and Tokyo 2021 Olympics. Though they cover serveral typical patterns of different social events, the number of events is somehow limited. While in real-world, there are lots of  different events. In that sense, the proposed event-dataset and method is somehow limited. How can the dataset  and the model scale up w.r.t the event types?

(2) LLM-based methods can have different output with the same input, it would be better to report the results across different runs to see whether the performance is stable or consitent.

(3) What is the running time at the inference time, since the model requires multiple LLM runs? which could cost a lot time and tokens, especially when simulation for a mega-city is required.

**Questions:**

Please see the weakness above.

---

> ### Author Response · Authors · 2025-11-21
> **Clarifications to Reviewer k5vy**
>
> We sincerely thank the reviewer k5vy for highly insightful and constructive comments.
>
> **Question1:** One concern is number of events in the paper, which only covers Typhoon, COVID-19 Pandemic, and Tokyo 2021 Olympics. Though they cover serveral typical patterns of different social events, the number of events is somehow limited. While in real-world, there are lots of different events. In that sense, the proposed event-dataset and method is somehow limited. How can the dataset and the model scale up w.r.t the event types?
>
> **Answer1:**
>
> Regarding dataset scalability, our collection pipeline is automated and event-agnostic. It can scale to new event types, implying a low-cost query of data using different time windows rather than constructing new datasets from scratch.
>
> Regarding model scalability, the framework is similarly event-agnostic by design.
> The Event Schema module standardizes diverse narratives into a unified representation, while the FTT-based Self-Alignment applies a universal conflict-resolution logic to generalize across unseen events.
>
> Experimentally, ELLMob has achieved consistent advanced performance across three types of events, validating its extendibility. We further confirmed this by evaluating on a 'New Year' scenario, where ELLMob consistently achieves advanced performance (see *Global Clarification E6*, at the top of this page).
>
> ---
>
> **Question2:** LLM-based methods can have different output with the same input, it would be better to report the results across different runs to see whether the performance is stable or consitent.
>
> **Answer2:**
>
> To verify robustness, we conducted repeated experiments across 5 random seeds as *Global Clarification E1* (at the top of this page) shown, observing small standard deviations that confirm ELLMob's statistical stability.
>
> ---
>
> **Question3:** What is the running time at the inference time, since the model requires multiple LLM runs? which could cost a lot time and tokens, especially when simulation for a mega-city is required.
>
> **Answer3:**
>
> We assessed computational cost on a “per person per day” basis as *Global Clarification E2* (at the top of this page) shown.
> Although ELLMob introduces additional overhead compared to single-pass models, it remains more efficient than generic reflection baselines, with identical reflection iterations for a fair comparison.
>
> This efficiency stems from our structured "Gist-based" alignment (FTT theory), which provides targeted guidance to converge rapidly, avoiding the excessive token consumption and latency inherent in open-ended iterative refinement.
> Simultaneously, ELLMob achieves advanced performance across all metrics, demonstrating optimal cost-effectiveness.
>
> Each user's mobility can be generated in parallel, enabling city-scale simulation.

---

> > ### Comment · Reviewer_k5vy · 2025-11-22
> >
> > Thanks for the authors' response. My concern still remains how the model can scale w.r.t the event types, in the new experiments, it shows "evaluating on a 'New Year' scenario", not when many new types of events shows

---

> ### Author Response · Authors · 2025-11-23
>
> We thank reviewer k5vy for the further question.
>
>
> We would clarify that our scalability comes from generalization across distinct event types. The Event Schema module converts different event-type narratives into standardized semantic descriptors (e.g., traffic impact). A "new event type" is a combination of these descriptors, which the LLM can process. The FTT-based Self-Alignment is a universal conflict-resolution logic and can generalize across event types.
> ​
>
> To verify that this generalization works, selected 4 events represent diverse and contrasting types. Importantly, we used the identical framework and parameters for all events:
> ​
>
> - Restrictive vs. Promotive: COVID-19/Typhoon restricts movement, while the New Year promotes gathering.
> ​
>
> - Hybrid Restrictive and Promotive: 2021 Tokyo Olympics (under COVID-19)
> ​
>
> - Stochastic vs. Periodic: Typhoon is unpredictable/sudden, while New Year's is periodic/cyclic.
> ​
>
> - Local vs. Global: 2021 Tokyo Olympics is concentrated in specific areas, while COVID-19 affects the entire city.
> ​
>
> The consistent performance across diverse event types confirms that the model is not overfitting to specific event types, demonstrating its potential to generalize to unseen event types.
> ​
>
> Finally, we respectfully note that constructing high-granularity & many types covered event-mobility datasets remains a challenge for the entire research community due to sparse event data and privacy constraints.
>
>
> To the best of our knowledge, our work is the first attempt that covers a wide spectrum of distinct event types (long-term, short-term, and different event semantics), providing representative scalability validation. A summary table of event types will be added in the revised version.
>
> ​
> While our framework is theoretically capable of scaling to any event types, extensive empirical validation is currently constrained by the scarcity of high-quality ground truth data. We hope that this dataset and framework can stimulate further research in event-driven mobility studies.

---

> > ### Comment · Reviewer_k5vy · 2025-11-23
> >
> > Thanks for addressing my concerns, I don't have any questions right now.

---

> ### Author Response · Authors · 2025-11-23
>
> We sincerely appreciate your constructive comments and are pleased that your concerns have been addressed.
> Your feedback has been invaluable in helping improve our work.

---

### Official Review · Reviewer_3JM6 · 2025-10-22

**Soundness:** 2
**Presentation:** 3
**Contribution:** 2
**Rating:** 4
**Confidence:** 4

**Summary:**

This paper introduces ELLMob, a self-aligned LLM framework for event-driven human mobility generation. It highlights two main gaps in current work: (1) the lack of event-annotated mobility datasets, and (2) the inability of LLM-based trajectory generators to reconcile habitual patterns with event-induced constraints. To address these, the authors construct a Tokyo-based event-annotated dataset (Typhoon Hagibis, COVID-19, Tokyo 2021 Olympics) and propose a Fuzzy-Trace Theory (FTT)-inspired framework that extracts three “gists” (pattern, event, action) and performs iterative “reflection-based alignment” to generate plausible trajectories.

**Strengths:**

Strength:
S1: The “event-driven mobility” setup moves beyond routine trajectory generation and has real-world applications in urban crisis modeling and planning.
S2: The constructed event-annotated dataset is potentially useful for studying mobility under disruptions. The data collection and anonymization steps are clearly documented.

**Weaknesses:**

Weakness:
W1: The core of ELLMob lies in a prompt-based reflection loop with heuristic “alignment auditing.” While reflection-based prompting has been extensively studied in the LLM literature, this work does not introduce any new architecture or learning algorithm. Essentially, the contribution remains at the level of prompt engineering rather than methodological contribution and is quite limited.

W2: Although the proposed dataset includes event annotations (e.g., typhoon, pandemic, Olympics), the data collection process appears to cover only trajectory information, and the event annotation itself seems rather trivial. In this sense, many comparable datasets are already publicly available. Meanwhile, the dataset’s scale is small (only 1,100 users), and it is restricted to the Tokyo area, which further limits its generalizability and practical utility.

W3: Although the paper claims to be “theory-driven,” the use of FTT is largely metaphorical. The separation of gist into pattern-based and event-based forms is very intuitive for this task, but not necessarily derived from the cognitive theory itself. The attributes used to analyze trajectories as “pattern gist” or “event gist” are also intuitive and do not require FTT for guideline or serving as a core component. Meanwhile, it remains unclear for me how FTT concretely improves the model’s reasoning or generation performance.

W4: Baseline choice. It is unclear how event information is incorporated into the baseline methods. In addition, all LLM-based baselines are single-pass models without reflection or iterative refinement. Given that reflection mechanisms are now common in the LLM prompting literature, the comparison should include other iterative-reflection baselines.

W5: Backbone choice. All modules rely on proprietary models such as GPT-4 or Gemini for prompting (w/o any open-source LLMs). The reflection loop is not quantified in terms of computational cost or token usage. Reproducibility might be limited due to closed-source LLM dependencies.

**Questions:**

Please refer to weakness.

---

> ### Author Response · Authors · 2025-11-24
> **Clarifications to Reviewer 3JM6**
>
> We thank the reviewer 3JM6 for the time and effort devoted to reviewing our manuscript and providing helpful suggestions.
>
> **Question1:** The core of ELLMob lies in a prompt-based reflection loop with heuristic “alignment auditing.” While reflection-based prompting has been extensively studied in the LLM literature, this work does not introduce any new architecture or learning algorithm. Essentially, the contribution remains at the level of prompt engineering rather than methodological contribution and is quite limited.
>
> **Answer1:**
>
> We would like to clarify that, beyond generic prompt-based reflection, we propose a cognitive theory–based reflection mechanism, bridging cognitive science and mobility modeling.
>
> ### Cognitive theory–based reflection (ours) vs. generic prompt-based reflection
> At the representation level, FTT-inspired reflection mimics human decision-making by *explicitly constructing structured decision variables (gists)*, whereas baselines such as Reflexion [1] *operate directly on unstructured natural-language trajectories* and thus struggle to disentangle the underlying decision drivers, often resulting in surface-level corrections rather than cognitively grounded behavioral adjustments.
> At the decision level, FTT-inspired reflection *treats pattern gists and event gists as competing qualitative options and revises mobility via dual-axis alignment* until this competition is resolved. In contrast, generic reflection methods such as Reflexion [1] simply *ask the model to “improve” its previous answer without explicitly modeling the pattern–event trade-off*, which easily yields trajectories that appear locally reasonable but either violate event constraints or unnecessarily disrupt routine behavior.
>
> ### Performance compared with generic prompt-based reflection
> Our experiments demonstrate that this theory-driven reflection consistently outperforms standard reflection loops on event-driven mobility generation tasks (see *Global Clarification E4*, at the top of this page).
>
> We will clarify this positioning and better distinguish our work from generic reflection-based prompting in the revision.
>
> ---
>
> **Reference**
>
> [1] Reflexion: Language Agents with Verbal Reinforcement Learning

---

> ### Author Response · Authors · 2025-11-24
>
> **Question2:** Although the proposed dataset includes event annotations (e.g., typhoon, pandemic, Olympics), the data collection process appears to cover only trajectory information, and the event annotation itself seems rather trivial. In this sense, many comparable datasets are already publicly available. Meanwhile, the dataset’s scale is small (only 1,100 users), and it is restricted to the Tokyo area, which further limits its generalizability and practical utility.
>
> **Answer2:**
>
> ### Continuity and integrity beyond simple event labels
> The key value of our dataset *lies in the temporal continuity and integrity of user trajectories, specifically within event windows, rather than the mere inclusion of event labels*.
> While identifying event timelines is straightforward, capturing high-granularity mobility records that strictly align with these periods is challenging due to data sparsity.
> Most comparable datasets (e.g., Foursquare [1], GeoLife [2]) contain sparse records that often encompass only routine periods, leaving sparse data during significant societal shifts.
> In contrast, we rigorously screened 1,100 users to *ensure they possess robust historical information and traceable mobility records spanning the 2019–2021 timeline*.
> This rigorous filtering ensures that:
> - Continuity across Phases: Users have continuous records covering the pre-event and event stage,  (e.g., tracking the complete transition from normalcy to the COVID-19 state of emergency).
> - Event Responsiveness: The dataset captures behavioral changes during short-term and non-routine events (e.g., Tokyo New Year, Olympics) which are typically missed in sparse datasets.
> - Our selection process preserves the real-world power-law distribution (Figure A2) while *guaranteeing that the events are backed by sufficient data density to support reliable causal observation*. For example, our data can trace a user who commutes daily to the city center before COVID-19 but shifts to short-distance grocery trips near home during emergency declarations, or whose outdoor activities sharply decrease on Typhoon Hagibis day compared with the surrounding days.
>
> ### Dataset scale
> In terms of scale, our dataset includes 1,100 users, exceeding representative mobility datasets such as Foursquare-NYC [1], and contains a larger volume of check-ins than SIN [3], GeoText [2], Foursquare-NYC [1], and Foursquare-TKY [1].  We further expanded the dataset with additional records covering the 2020 Tokyo New Year and mobility trajectories of COVID-19 from 1,100 users in Osaka.
>
> ### Geographic consistency and generalizability
> Focusing on the Tokyo metropolitan area ensures that they are subject to the same event conditions (e.g., policy interventions, infrastructure, weather system), thereby *reducing confounding factors introduced by geographic heterogeneity*. As for generalizability, ELLMOB maintains superior performance over all baselines on the Osaka dataset (see *Global Clarification E5*, at the top of this page) as well, highlighting the potential of this dataset to guide the design of generalizable frameworks.
>
> ### A new benchmark for event-driven mobility
>
> Current representative datasets (e.g., Foursquare-TKY[1] also collected in Tokyo) primarily capture routine patterns but fail to handle mobility reshaped by large-scale events. To the best of our knowledge, our dataset is the *first to cover a broad spectrum of distinct event types (long-term vs. short-term, diverse semantics) with continuous pre- and during-event trajectories*. This fills a gap in studying event-driven human mobility shifts.
>
> ---
>
> **Reference**
>
> [1] Modeling User Activity Preference by Leveraging User Spatial Temporal Characteristics in LBSNs.
>
> [2] GeoText: an intelligent dynamic geometry textbook
>
> [3] Time-aware point-of-interest recommendation
>
> [4] How do cities flow in an emergency? Tracing human mobility patterns during a natural disaster with big data and geospatial data science.

---

> ### Author Response · Authors · 2025-11-24
>
> **Question3:** Although the paper claims to be “theory-driven,” the use of FTT is largely metaphorical. The separation of gist into pattern-based and event-based forms is very intuitive for this task, but not necessarily derived from the cognitive theory itself. The attributes used to analyze trajectories as “pattern gist” or “event gist” are also intuitive and do not require FTT for guideline or serving as a core component. Meanwhile, it remains unclear for me how FTT concretely improves the model’s reasoning or generation performance.
>
> **Answer3:**
>
> We appreciate the reviewer’s concern and agree that our design choices may look intuitive at first glance. However, *FTT itself is a cognitive theory of intuitive decision making*, and in our work it provides a principled basis for which structures to model and how to organize them. Without FTT, a simple “intuitive” design would more *likely collapse all information into a single, undifferentiated context representation*.
>
> ### Multi-gist decision vs. single undifferentiated context decision
> First, FTT holds that people can form multiple gists of the same situation and rely on these gist representations when making decisions, which motivates us to build a multi-gist decision framework. The pattern-based and event-based gists are task-specific instantiations of these gist perspectives. *Without FTT, there is no principled reason to explicitly model two interacting gist types rather than, for example, a single undifferentiated context vector*.
>
> ### Unified gist space vs. heterogeneous verbatim inputs
> Second, in our setting, historical trajectories, event descriptions, and candidate mobility are all verbatim data (visited locations, latitude–longitude coordinates, free-form text) that live in heterogeneous spaces. *FTT’s distinction between verbatim and gist representations guides us to map these heterogeneous inputs into a shared gist space*, in which internal (pattern–event) and external (event–action) alignment can be performed in a consistent way.
>
>
> ### Interpretable bottom-line attributes vs. arbitrary feature engineering
> Third, FTT characterizes gists as bottom-line meanings: coarse, categorical, and interpretable. This principle drives our choice to represent pattern, event, and action gists using a small set of human-interpretable attributes (e.g., Core Behavior, Points of Inertia, and other attributes in Table 4) instead of raw trajectory statistics. We agree that these attributes are intuitive for the task, but their coarse, categorical, bottom-line formulation and their organization into pattern/event/action gists are *directly motivated by FTT, rather than being arbitrary feature engineering*.
>
> ### Ablation evidence of FTT’s concrete impact
> To address the reviewer’s concern about concrete impact, we include ablations that isolate FTT-style design choices. Variants that rely mainly on verbatim features (w/ verbatim) or remove the bottom-line gist component (w/o bottom) consistently perform worse than ELLMob across different events. These results indicate that the FTT-guided framework design concretely improves generation quality under event-driven mobility.
>
>
> *Typhoon Hagibis*
>
> | Model       | SI     | SD     | CD     | SGD    |
> |------------|--------|--------|--------|--------|
> | w/ verbatim | 0.1511 | 0.0724 | 0.0230 | 0.0595 |
> | w/o bottom | 0.0978 | 0.0443 | 0.0101 | 0.0327 |
> | ELLMob     | 0.0642 | 0.0200 | 0.0041 | 0.0173 |
>
>
> *COVID-19 Pandemic*
>
> | Model       | SI     | SD     | CD     | SGD    |
> |------------|--------|--------|--------|--------|
> | w/ verbatim | 0.2071 | 0.0713 | 0.0324 | 0.0422 |
> | w/o bottom | 0.1687 | 0.0690 | 0.0305 | 0.0333 |
> | ELLMob     | 0.1003 | 0.0444 | 0.0080 | 0.0268 |
>
>
>
> *Tokyo 2021 Olympics*
> | Model        | SI     | SD     | CD     | SGD    |
> |-------------|--------|--------|--------|--------|
> | w/ verbatim | 0.1030 | 0.0452 | 0.0054 | 0.0189 |
> | w/o bottom  | 0.0892 | 0.0364 | 0.0036 | 0.0082 |
> | ELLMob      | 0.0617 | 0.0061 | 0.0022 | 0.0035 |

---

> ### Author Response · Authors · 2025-11-24
>
> **Question4:** Baseline choice. It is unclear how event information is incorporated into the baseline methods (1). In addition, all LLM-based baselines are single-pass models without reflection or iterative refinement. Given that reflection mechanisms are now common in the LLM prompting literature, the comparison should include other iterative-reflection baselines (2).
>
> **Answer4.1:**
>
> Event context is integrated into the prompts of all LLM-based baselines. Specifically, event descriptions (type, time, location, constraints) are provided in natural language at the beginning of each prompt.
>
> **Answer4.2:**
>
> We compared ELLMob against representative iterative-reflection baselines (Reflexion, SELF-REFINE, Air) with consistent settings (see *Global Clarification E4*, at the top of this page) .
> Experiments confirm that ELLMob achieves the best performance, demonstrating its advance.
>
> ---
> **Question5:** Backbone choice. All modules rely on proprietary models such as GPT-4 or Gemini for prompting (w/o any open-source LLMs). The reflection loop is not quantified in terms of computational cost or token usage (1). Reproducibility might be limited due to closed-source LLM dependencies (2).
>
> **Answer5.1:**
>
> We compared ELLMob against representative iterative-reflection baselines with consistent settings.
> Specifically, we evaluated generation quality using SI, SD, CD, and SGD (see *Global Clarification E4*, at the top of this page), and assessed efficiency using Token Count, Inference Time, and Cost (see *Global Clarification E2*, at the top of this page). *Experiments confirm that ELLMob achieves the best performance with the lowest computational overhead*, demonstrating its efficiency including faster convergence.
>
> **Answer5.2:**
>
> We validated ELLMob on open-source models (Llama-3-8B, Qwen-2.5-14B, DeepSeek-R1-Distill-Qwen-7B), confirming its robustness (see *Global Clarification E3*, at the top of this page).

---

> > ### Author Response · Authors · 2025-11-27
> >
> > As the deadline for the discussion period is approaching, we wanted to check if our response has sufficiently addressed your concerns. If there are any remaining questions, please let us know so we can provide further clarification.

---

### Official Review · Reviewer_bqV1 · 2025-10-26

**Soundness:** 3
**Presentation:** 3
**Contribution:** 3
**Rating:** 6
**Confidence:** 4

**Summary:**

This paper introduces the task of event-driven human mobility generation. To address the shortcomings of existing methods under atypical scenarios, which tend either to replay routine travel or to over-conform to event constraints, this paper constructs an explicitly event-annotated dataset covering three Tokyo events plus a normal period, and proposes ELLMob, a cognition-theory–driven framework. ELLMob first distills long-form news and policy text into a structured event context, then extracts three kinds of gists (pattern, event, and action) and runs an iterative reflect–refine loop that conducts dual auditing for internal alignment  and external alignment, until a trajectory is produced that is both habitually grounded and event-compliant. Using Jensen–Shannon Divergence (JSD) on the distributions of Step Interval (SI), Step Distance (SD), Category Distribution (CD), and Spatial Grid Distribution (SGD) for evaluation, ELLMob significantly outperforms strong baselines across all three events, and ablations confirm that the self-alignment mechanism is the key source of improvement.

**Strengths:**

(1) This paper introduces the first event-centric, fully featured, explicitly annotated dataset, providing a solid foundation for studying non-routine deviations in mobility.
(2) This paper converts event understanding from free text into a structured event context directly consumable by models, reducing loss of critical information.
(3) This paper proposes the ELLMob framework, which uses gist-based alignment to explicitly reconcile competing mobility decisions.
(4) ELLMob achieves significant improvements over existing methods.

**Weaknesses:**

(1) The data come exclusively from Twitter/Foursquare, which may skew the sample toward younger, more homogeneous users, introducing selection and behavioral biases.
(2) The paper lacks details about the LLM type and hyperparameters; providing a fuller description would improve clarity and reproducibility.
(3) For the reflect–regenerate procedure, if constraints remain unmet after K iterations, please offer a further remedy or fallback strategy.

**Questions:**

(1) The study’s geographic and event scope is relatively narrow, which focused solely on Tokyo and three event types within Tokyo, does it have a certain degree of transferability?
(2) The paper does not discuss how the choice of LLM might affect the method’s performance, does model selection materially influence results?

---

> ### Author Response · Authors · 2025-11-22
> **Clarifications to Reviewer bqV1**
>
> We sincerely appreciate the reviewer bqV1’s valuable suggestions.
>
> **Question1:** The data come exclusively from Twitter/Foursquare, which may skew the sample toward younger, more homogeneous users, introducing selection and behavioral biases.
>
> **Answer1:**
>
> We acknowledge that Twitter/Foursquare data has demographic biases.
> This is a pervasive limitation inherent to social check-in data, commonly observed in LBSN research [1, 2, 3, 4].
> We will document this constraint in the revision.
> In future work, we plan to extend our evaluation to more diverse data sources to validate the model's generalizability across broader population groups.
>
> ---
>
> **Question2:** The paper lacks details about the LLM type and hyperparameters; providing a fuller description would improve clarity and reproducibility.
>
> **Answer2:**
>
> In our paper, the key parameters are detailed in Section 6.1 (Implementation Details). For completeness, we also note that Top-p is set to 1 and the GPT4o-mini model (version 2025-01-01-preview) is used. Please feel free to let us know if any additional parameters would be helpful.
>
> ---
>
> **Question3:** For the reflect–regenerate procedure, if constraints remain unmet after K iterations, please offer a further remedy or fallback strategy.
>
>
> **Answer3:**
>
> Based on our empirical observations, scenarios where constraints remain unmet after $K$ iterations are rare. For these exceptional cases, our fallback strategy is to adopt the last planned trajectory generated in the final iteration. As described at the end of the "Corrective refinement" paragraph in Section 5.2.2.
>
> ---
>
> **Question4:** The study’s geographic and event scope is relatively narrow, which focused solely on Tokyo and three event types within Tokyo, does it have a certain degree of transferability?
>
> **Answer4:**
>
> Regarding our framework, the Event Schema module standardizes diverse event narratives into a unified representation and the FTT theory-based Self-Alignment module applies a universal conflict-resolution logic to generalize across unseen events.
>
> We further demonstrate spatial and event transferability by achieving superior performance over the baselines on the newly introduced Osaka dataset (COVID-19) and the New Year scenario (see *Global Clarification E5 and E6*, at the top of this page).
>
> ---
>
> **Question5:** The paper does not discuss how the choice of LLM might affect the method’s performance, does model selection materially influence results?
>
>
> **Answer5:**
> We previously analyzed model adaptability in Appendix G using Gemini Flash 2.0, and have now extended this evaluation to open-source models (Llama-3-8B, Qwen-2.5-14B, DeepSeek-R1-Distill-Qwen-7B) (see *Global Clarification E3*, at the top of this page). The results confirm that ELLMob remains robust across diverse model architectures.
>
> ---
>
> **Reference**
>
> [1] Massive-STEPS: Massive Semantic Trajectories for Understanding POI Check-ins--Dataset and Benchmarks (Massive-STEPS dataset)
>
> [2] Participatory cultural mapping based on collective behavior data in location-based social networks (Foursquare-Global dataset)
>
> [3] A latent variable model for geographic lexical variation (GeoText dataset)
>
> [4] Delineation of an urban agglomeration boundary based on Sina Weibo microblog ‘check-in’data: A case study of the Yangtze River Delta

---

> > ### Author Response · Authors · 2025-11-27
> >
> > As the deadline for the discussion period is approaching, we wanted to check if our response has sufficiently addressed your concerns. If there are any remaining questions, please let us know so we can provide further clarification.

---

### Official Review · Reviewer_WGcR · 2025-10-26

**Soundness:** 3
**Presentation:** 3
**Contribution:** 3
**Rating:** 6
**Confidence:** 2

**Summary:**

The paper introduces ELLMob, an event-driven human-mobility generator that reconciles a user’s habitual patterns with exogenous event constraints via a self-alignment loop. It also provides a new event-annotated mobility dataset for Tokyo spanning three events. Evaluation shows ELLMob outperforms baselines on the dataset.

**Strengths:**

-paper is overall well-written and easy to follow

-new dataset with event context has been provided

-the proposed method ELLMob is neat and shows consistent improvements across SI/SD/CD/SGD over the baseline methods. The ablation study shows both the schema and the reflection matter.

**Weaknesses:**

-methodology contribution is relatively on the incremental side. The contribution is more on applying the concept of self-alignment along with some domain specific heuristic into the human-mobility prediction task.

-evaluation can be more comprehensive:
(1) variance of results are absent
(2) all scenarios are single-city (Tokyo)

**Questions:**

Can you compare the token usage / latency of ELLMob and the baselines?

---

> ### Author Response · Authors · 2025-11-23
> **Clarifications to Reviewer WGcR**
>
> We sincerely thank the reviewer WGcR for these thoughtful comments and important questions.
>
> **Question1:** Methodology contribution is relatively on the incremental side. The contribution is more on applying the concept of self-alignment along with some domain specific heuristic into the human-mobility prediction task.
>
> **Answer1:**
>
> In prior works, generic self-alignment is typically framed as a tool for error correction, for example to mitigate hallucinations or logical inconsistencies.
> In contrast, ELLMob addresses a decision-making dilemma in event-driven human mobility: the tension between habitual inertia and event-induced constraints. Our proposed conflict-reconciliation self-alignment mechanism is designed to arbitrate this duality.
>
> Our framework is theory-heuristic and inspired by Fuzzy-Trace Theory, transforming mobility generation into interpretable cognitive reasoning.
> This allows the model to adaptively balance habit and event constraints, mimicking human decision-making instead of rigidly following domain rules.
>
> Empirically, removing this module results in a substantial performance drop (e.g., 57% in COVID-19 SI), and replacing it with generic self-alignment methods also degrades performance (see *Global Clarification E4*, at the top of this page), underscoring its value in event-driven mobility generation.
>
> ---
>
> **Question2:** Evaluation can be more comprehensive: (1) variance of results are absent (2) all scenarios are single-city (Tokyo)
>
> **Answer2.1:**
>
> To verify robustness, we conducted repeated experiments across 5 random seeds (see *Global Clarification E1*, at the top of this page), observing small standard deviations that confirm ELLMob's statistical stability.
>
> **Answer2.2:**
>
> To demonstrate spatial generalizability, we extended evaluations to Osaka using a new dataset of 1,100 users (matching the Tokyo scale) during the COVID-19 pandemic (see *Global Clarification E5*, at the top of this page).
> ELLMob consistently outperforms baselines in this new geography, proving its adaptability to diverse urban layouts beyond Tokyo.
>
> ---
>
> **Question3:** Can you compare the token usage / latency of ELLMob and the baselines?
>
> **Answer3:**
>
> We assessed computational cost on a "per person per day" basis (see *Global Clarification E2*, at the top of this page).
> Although ELLMob introduces additional overhead compared to single-pass models, it remains more efficient than generic reflection baselines, with identical reflection iterations for a fair comparison.
> This efficiency stems from our structured "Gist-based" alignment (FTT theory), which provides targeted guidance to converge rapidly, avoiding the excessive token consumption and latency inherent in open-ended iterative refinement.
> Also, ELLMob achieves advanced performance across all metrics, demonstrating optimal cost-effectiveness.

---

> > ### Author Response · Authors · 2025-11-27
> >
> > As the deadline for the discussion period is approaching, we wanted to check if our response has sufficiently addressed your concerns. If there are any remaining questions, please let us know so we can provide further clarification.

---

### Author Response · Authors · 2025-11-21
**Global clarification.**

We sincerely thank all reviewers for their insightful and valuable suggestions. In response, we provide additional experiments and analyses as a global clarification to further address the raised concerns.

**E1: Model Stability Analysis**

*Typhoon Hagibis*
| Model    | SI | SD | CD | SGD |
|----------|----|----|----|-----|
| LLMOB    | 0.1057 ± 0.0277 | 0.1509 ± 0.0386 | 0.0497 ± 0.0243 | 0.0319 ± 0.0170 |
| LLM-MOB  | 0.1501 ± 0.0170 | 0.0558 ± 0.0208 | 0.0352 ± 0.0096 | 0.0532 ± 0.0430 |
| LLM-MOVE | 0.1242 ± 0.0035 | 0.0407 ± 0.0021 | 0.0131 ± 0.0007 | 0.0290 ± 0.0014 |
| LLM-ZS   | 0.1601 ± 0.0071 | 0.1335 ± 0.0055 | 0.0155 ± 0.0008 | 0.0750 ± 0.0025 |
| ELLMob   | 0.0649 ± 0.0030 | 0.0220 ± 0.0044 | 0.0046 ± 0.0012 | 0.0162 ± 0.0018 |

*COVID-19 Pandemic*
| Model    | SI | SD | CD | SGD |
|----------|----|----|----|-----|
| LLMOB    | 0.1128 ± 0.0173 | 0.1221 ± 0.0298 | 0.0158 ± 0.0038 | 0.0295 ± 0.0047 |
| LLM-MOB  | 0.1099 ± 0.0107 | 0.0613 ± 0.0263 | 0.0191 ± 0.0072 | 0.0359 ± 0.0066 |
| LLM-MOVE | 0.1789 ± 0.0241 | 0.0511 ± 0.0044 | 0.0457 ± 0.0185 | 0.0544 ± 0.0067 |
| LLM-ZS   | 0.1108 ± 0.0103 | 0.0551 ± 0.0025 | 0.0442 ± 0.0074 | 0.0605 ± 0.0056 |
| ELLMob   | 0.1002 ± 0.0094 | 0.0443 ± 0.0037 | 0.0079 ± 0.0015 | 0.0279 ± 0.0027 |

*Tokyo 2021 Olympics*
| Model    | SI | SD | CD | SGD |
|----------|----|----|----|-----|
| LLMOB    | 0.1042 ± 0.0131 | 0.0291 ± 0.0036 | 0.0115 ± 0.0017 | 0.0054 ± 0.0006 |
| LLM-MOB  | 0.1123 ± 0.0067 | 0.0338 ± 0.0083 | 0.0157 ± 0.0084 | 0.0172 ± 0.0127 |
| LLM-MOVE | 0.2027 ± 0.0050 | 0.0304 ± 0.0014 | 0.0111 ± 0.0009 | 0.0053 ± 0.0004 |
| LLM-ZS   | 0.0953 ± 0.0021 | 0.0320 ± 0.0014 | 0.0147 ± 0.0021 | 0.0060 ± 0.0011 |
| ELLMob   | 0.0606 ± 0.0016 | 0.0063 ± 0.0002 | 0.0021 ± 0.0002 | 0.0037 ± 0.0002 |

---

**E2: Computational Cost (Token / Latency) Analysis**

Costs are calculated using GPT4o-mini API pricing.

| Model        | Token Count (per person per day) | Inference Time (seconds) / (per person per day) | Cost (dollars) / (per person per day) |
|-------------|-----------------------------|-----------------|----------------------------|
| LLMOB       | 1,271  | 10.12 | 0.00030 |
| LLM-MOB     | 3,954  | 3.72  | 0.00064 |
| LLM-MOVE    | 4,954  | 4.05  | 0.00078 |
| LLM-ZS      | 5,184  | 3.34  | 0.00080 |
| Reflexion [1]   | 26,057 | 27.12 | 0.00417 |
| SELF-REFINE [2] | 15,382 | 21.16 | 0.00258 |
| Air [3]         | 15,514 | 20.50 | 0.00260 |
| ELLMob  | 9,569  | 18.68 | 0.00170 |

---

**E3: LLM Backbone Ablation Study**

*Typhoon Hagibis*
| Model                 | SI     | SD     | CD     | SGD    |
|----------------------|--------|--------|--------|--------|
| Llama-3-8B            | 0.0663 | 0.0512 | 0.0004 | 0.0132 |
| Qwen-2.5-14B         | 0.1594 | 0.0607 | 0.0091 | 0.0115 |
| DeepSeek-R1-Distill-Qwen-7B   | 0.0881 | 0.0516 | 0.0033 | 0.0203 |
| GPT4o-mini   | 0.0642 | 0.0200 | 0.0041 | 0.0173 |
| Gemini Flash 2.0   | 0.0850 | 0.0267 | 0.0087 | 0.0160 |

*COVID-19 Pandemic*
| Model                 | SI     | SD     | CD     | SGD    |
|----------------------|--------|--------|--------|--------|
| Llama-3-8B            | 0.0669 | 0.0624 | 0.0087 | 0.0263 |
| Qwen-2.5-14B         | 0.0836 | 0.0646 | 0.0016 | 0.0225 |
| DeepSeek-R1-Distill-Qwen-7B   | 0.1104 | 0.0993 | 0.0163 | 0.0392 |
| GPT4o-mini   | 0.1003 | 0.0444 | 0.0080 | 0.0268 |
| Gemini Flash 2.0   | 0.0546 | 0.0586 | 0.0069 | 0.0210 |

*Tokyo 2021 Olympics*
| Model                 | SI     | SD     | CD     | SGD    |
|----------------------|--------|--------|--------|--------|
| Llama-3-8B            | 0.0407 | 0.0322 | 0.0023 | 0.0030 |
| Qwen-2.5-14B         | 0.0530 | 0.0238 | 0.0035 | 0.0021 |
| DeepSeek-R1-Distill-Qwen-7B   | 0.0570 | 0.0251 | 0.0015 | 0.0071 |
| GPT4o-mini   | 0.0617 | 0.0061 | 0.0022 | 0.0035 |
| Gemini Flash 2.0   | 0.0113 | 0.0074 | 0.0016 | 0.0048 |

---

**E4: Baseline Reflection Mechanism**

Here we select three representative reflection baselines [1,2,3] for comparison.

*Typhoon Hagibis*
| Model        | SI     | SD     | CD     | SGD    |
|-------------|--------|--------|--------|--------|
| Reflexion   | 0.1106 | 0.1282 | 0.0841 | 0.0855 |
| SELF-REFINE | 0.1979 | 0.0637 | 0.0135 | 0.0193 |
| Air         | 0.0764 | 0.0710 | 0.0198 | 0.0204 |
| ELLMob      | 0.0642 | 0.0200 | 0.0041 | 0.0173 |

*COVID-19 Pandemic*
| Model        | SI     | SD     | CD     | SGD    |
|-------------|--------|--------|--------|--------|
| Reflexion   | 0.1685 | 0.0588 | 0.0146 | 0.0269 |
| SELF-REFINE | 0.2122 | 0.1053 | 0.0344 | 0.0294 |
| Air         | 0.1858 | 0.0454 | 0.0256 | 0.0291 |
| ELLMob      | 0.1003 | 0.0444 | 0.0080 | 0.0268 |

*Tokyo 2021 Olympics*
| Model        | SI     | SD     | CD     | SGD    |
|-------------|--------|--------|--------|--------|
| Reflexion   | 0.1741 | 0.0308 | 0.0378 | 0.0704 |
| SELF-REFINE | 0.0826 | 0.0320 | 0.0073 | 0.0058 |
| Air         | 0.0774 | 0.0441 | 0.0053 | 0.0035 |
| ELLMob      | 0.0617 | 0.0061 | 0.0022 | 0.0035 |

---

> ### Author Response · Authors · 2025-11-21
> **Continue**
>
> **E5: Regional Restriction**
>
> Here we provide Osaka as a  case for regional restriction analysis.
>
> *COVID-19 Pandemic (Osaka)*
> | Model     | SI     | SD     | CD     | SGD    |
> |-----------|--------|--------|--------|--------|
> | LLMOB     | 0.1934 | 0.1589 | 0.0344 | 0.1201 |
> | LLM-MOB   | 0.1732 | 0.1617 | 0.0304 | 0.0988 |
> | LLM-MOVE  | 0.2134 | 0.1555 | 0.0698 | 0.1227 |
> | LLM-ZS    | 0.1531 | 0.1788 | 0.0299 | 0.1161 |
> | ELLMob | **0.1131** | **0.1001** | **0.0120** | **0.0556** |
>
> ---
>
> **E6: Event Restriction**
>
> *New Year*
>
> | Model    | SI     | SD     | CD     | SGD    |
> |----------|--------|--------|--------|--------|
> | LLMOB    | 0.0776 | 0.0391 | 0.0230 | 0.0422 |
> | LLM-MOB  | 0.1061 | 0.0413 | 0.0318 | 0.0379 |
> | LLM-MOVE | 0.2248 | 0.0493 | 0.0272 | 0.0550 |
> | LLM-ZS   | 0.0996 | 0.0497 | 0.0230 | 0.0481 |
> | ELLMob | **0.0598** | **0.0250** | **0.0200** | **0.0317** |
>
> ---
>
> **Reference**
>
> [1] Reflexion: Language Agents with Verbal Reinforcement Learning
>
> [2] SELF-REFINE: iterative refinement with self-feedback
>
> [3] AIR: Complex Instruction Generation via Automatic Iterative Refinement
>
> ---
>
> *Please let us know if any further information is needed.*

---

### Author Response · Authors · 2025-11-28
**Record of Reviewer Engagement and Score Updates (as of Nov. 29)**

Dear all,

We are sorry to hear about the recent OpenReview system issue, and we fully support the remedial actions.

To facilitate the final decision-making process, we provide a summary of the score changes and reviewer engagement status.

| Reviewer | Score Change | Status during Rebuttal |
| :--- | :--- | :--- |
| WGcR | 6 -> 6 (Unchanged) | No post-rebuttal engagement |
| bqV1 | 6 -> 6 (Unchanged) | No post-rebuttal engagement |
| 3JM6 | 4 -> 4 (Unchanged) | No post-rebuttal engagement |
| k5vy | 4 -> 6 | Concerns resolved. Stated: *"Thanks for addressing my concerns, I don't have any questions right now."* |
| **Average** | **5.0 -> 5.5** | |

**Timeline Clarification:** This score update was completed on **Nov. 23 at 03:51 AoE**, which was prior to the reported system issue.

We appreciate the time and effort the reviewers have dedicated to our work.

Best regards,

Authors

---

### Author Response · Authors · 2025-12-02
**Author Remarks to the Area Chair**

To facilitate your final assessment and minimize your workload, we provide a recap of paper's core storyline, a  summary of rebuttal outcomes, and a guide to our revisions.

# The Core Storyline

**tl;dr**
We study human mobility generation under large-scale societal events.

To address the data scarcity issue in human mobility research, we construct an **event-annotated** mobility dataset covering three major societal events (Typhoon Hagibis, COVID-19, and the Tokyo 2021 Olympics) with distinct event types (long-term, short-term, and different event semantics).

To tackle the challenge in event-driven human mobility generation, which has to reconcile competitions between users' habitual patterns and event-imposed constraints, we propose ELLMob, a **self-aligned** LLM framework based on the Fuzzy-Trace Theory.

# Summary of Review and Rebuttal Outcomes

Reviewers recognized:

- Modeling event-driven mobility is a **previously overlooked challenge** that moves beyond routine generation with significant real-world applications for urban crisis modeling.
- Our dataset fills a critical gap as the **first societal event-centric dataset** providing a solid foundation for studying non-routine mobility deviations under events.
- The proposed FTT-based framework is **technically sound and neat**, effectively reconciling competing mobility decisions and thereby outperforming all baselines.

In the rebuttal, we resolved common concerns by:

- **Clarifying Technical Contributions:** Distinguished our cognitive modeling from generic prompt engineering and proved superiority over reflection baselines (Reflexion, Self-Refine, Air).

- **Demonstrating Scalability:** Extended evaluation to a new city (Osaka) and a new event (New Year), supported by theoretical grounding on event types.

- **Verifying Stability and Efficiency:** Conducted stability analysis (5 seeds), open-source backbone testing (Llama-3-8B, Qwen-2.5-14B, and Deepseek-R1-Distill-Qwen-7B), and computational cost analysis.

While we haven't received the response from the other three reviewers, **Reviewer k5vy** confirmed that the **concerns had been resolved** ("Thanks for addressing my concerns, I don't have any questions right now.") and raised the score from 4 to 6 (timestamp: 03:51, Nov. 23rd, UTC-12 (AoE) timezone).

# Rebuttal and Revision Roadmap

We conducted a comprehensive revision of the entire manuscript to fully incorporate the constructive feedback from all reviewers.
All changes are highlighted in blue color.
Below is a roadmap mapping each reviewer's concerns to corresponding rebuttal and revised sections in paper.

**Reviewer 1 (WGcR)**

- *Weakness 1. (Incremental Design)*: [Rebuttal Answer1] Clarify methodological contribution in Section 1 (Introduction) Paragraph 4 and Section 5.2.2 (Paragraph 1).

- *Weakness 2.1. (Stability)*: [Rebuttal Answer2.1] Add stability analysis and extend evaluation in Appendix L.

- *Weakness 2.2. (Scalability)*: [Rebuttal Answer2.2] Add scalability analysis and extend evaluation in Section 6.2, and Appendix I.

- *Question 1. (Efficiency):* [Rebuttal Answer3] Add computational cost analysis in Section 6.6.

**Reviewer 2 (bqV1)**

- *Weakness 1. (Data Bias)*: [Rebuttal Answer1] Acknowledged demographic limitations as a pervasive constraint inherent to LBSN research in Section 7 (Conclusion).

- *Weakness 2. (Implementation Details)*: [Rebuttal Answer2] Further add model version and hyperparameter details in Section 6.1 (Implementation Details).

- *Weakness 3. (Fallback Strategy)*: [Rebuttal Answer3] Emphasize fallback mechanism in Section 5.2.2 (Corrective Refinement).

- *Question 1. (Scalability)*: [Rebuttal Answer4] Please refer to Reviewer 1 (WGcR), Weakness 2.2. and Appendix J.

- *Question 2. (LLM Backbone)*: [Rebuttal Answer5] Further included open-sourced model ablation study in Appendix H.

**Reviewer 3 (3JM6)**

- *Weakness 1. (Incremental Design)*: [Rebuttal Answer1] Clarify methodological contribution in Section 1 (Introduction)  Paragraph 4, Contribution, and add a comparative ablation in Section 6.5.

- *Weakness 2.*: [Rebuttal Answer2] Highlight dataset uniqueness in Section 4 (Paragraph 2).

- *Weakness 3. (FTT Theoretical Basis)*: [Rebuttal Answer3] Strengthen theoretical grounding in Section 5.2.1 and ablation study in Appendix F.

- *Weakness 4. (Baseline Setting)*: [Rebuttal Answer4.1-4.2] Clarify context setups in Section 6.1 (Baselines.) and add comparative ablation with iterative reflection baselines in Section 6.5.

- *Weakness 5. (LLM Backbone & Efficiency)*: [Rebuttal Answer5.1-5.2] Please refer to Reviewer 1 (WGcR), Question 1. and Reviewer 2 (bqV1) Question 2.

**Reviewer 3 (k5vy)**

- *Weakness 1. (Scalability)*: [Rebuttal Answer1] Please refer to Reviewer 2 (bqV1), Question 1.

- *Weakness 2. (Stability)*: [Rebuttal Answer2] Please refer to Reviewer 1 (WGcR), Weakness 2.1.

- *Weakness 3. (Efficiency)*: [Rebuttal Answer3] Please refer to Reviewer 1 (WGcR), Question 1.

---

### Meta-Review · Area_Chair_gYxF · 2025-12-13

**Summary:**

This work addresses two critical gaps in LLM-based human mobility generation: the lack of event-annotated datasets for non-routine mobility modeling and the inability to reconcile users’ habitual patterns with event-induced constraints. Reviewers raised concerns about incremental methodological contribution, dataset limitations, metaphorical FTT usage, and baseline gaps. The rebuttal resolved most empirical concerns raised by reviewers via additional experiments, though residual gaps (such as FTT’s theoretical formalization and dataset generalizability highlighted by Reviewer 3JM6) remain. Overall, ELLMob’s novel dataset and cognitively grounded conflict-reconciliation framework fill key gaps in event-driven mobility modeling, I lean toward acceptance.

**Reviewer Concerns:**

1. The remaining gap is theoretical formalization: FTT is still primarily a guiding principle rather than a rigorously derived, falsifiable cognitive model embedded in the method.

**Reviewer Scores:**

WGcR: 6 (Marginally above acceptance).
bqV1: 6 (Marginally above acceptance).
3JM6: 4 (Marginally below acceptance).
k5vy: 4 (Marginally below acceptance)-> 6 (Marginally above acceptance). Timeline Clarification: This score update was completed on Nov. 23 at 03:51 AoE, which was prior to the reported system issue.

---

### Decision · Program_Chairs · 2026-01-26

Accept (Poster)